# The impact of textual elements on the comprehensibility of drug label instructions (DLIs): A systematic review

Ekram Maghroudi[1,2]☯*, Charlotte Miriam Joyce van Hooijdonk[3]☯, Heidi van de Bruinhorst[1,4], Liset van Dijk[5,6], Jany Rademakers[2,5], Sander Diederik Borgsteede[1]

1 Department of Clinical Decision Support, Health Base Foundation, Houten, The Netherlands, 2 Department of Family Medicine, Maastricht University, CAPHRI, Maastricht, The Netherlands, 3 Faculty of Humanities, Department of Languages, Literature & Communication, Universiteit Utrecht, Utrecht, The Netherlands, 4 Universiteit Utrecht, Pharmacy, Utrecht, The Netherlands, 5 Nivel, Netherlands Institute for Health Services Research, Utrecht, The Netherlands, 6 Faculty of Mathematics and Natural Sciences, Department of PharmacoTherapy, -Epidemiology & -Economics (PTEE), Groningen Research Institute of Pharmacy, University of Groningen, Groningen, The Netherlands

☯ These authors contributed equally to this work.
* Ekram.maghroudi@healthbase.nl

**Data Availability Statement:** All relevant data are within the manuscript and its Supporting Information files.

## Abstract

### Introduction

Correct interpretation of drug labels instructions (DLIs) is needed for safe use and better adherence to prescribed drugs. DLIs are often too difficult for patients, especially for those with limited health literacy. What is yet unknown, is how specific textual elements in DLIs (e.g., the presentation of numbers, or use of medical jargon) and patients' health literacy skills are related to the comprehension of DLIs. In order to provide concrete directions for health professionals on how to optimize drug prescriptions, we performed a systematic review to summarize the available research findings on which textual elements facilitate or hinder the correct interpretation of DLIs in relation to patients' health literacy.

### Method

A systematic search was performed in PubMed, EMBASE, PsychINFO, and Smartcat (until April 2019) to identify studies investigating textual elements that facilitate or hinder the correct interpretation of DLIs in relation to patients' health literacy.

### Results

A total of 434 studies were identified of which 28 studies met our inclusion criteria. We found that textual elements contributing to the correct interpretation of DLIs were: using explicit time periods in dosage instructions, using plain language, presenting numbers in a numerical format, and providing DLIs in patients' native language. Multistep instructions per instruction line, using abbreviations and medical jargon seem to hinder the correct interpretation of DLIs. Although health literacy was taken into account in a majority of the studies, none of them assessed the effectiveness of specific textual elements on patients' comprehensibility of DLIs.

**Funding:** This work was supported by ZonMw, under Grant 848022004.

**Competing interests:** The authors have declared that no competing interests exist.

## Conclusion

Based on our findings, we provide an overview of textual elements that contribute to the correct interpretation of DLIs. Optimizing the textual instruction on drug labels may increase the safety and adherence to prescribed drugs, taking into account that a significant proportion of patients has low health literacy.

## Introduction

Instructions on how to use drugs are an essential part of patients' drug management. Patients' adherence to drug instructions influences the effectiveness of their therapy [1]. Patients receive information about their drug therapy from different sources, including information from the prescriber, information in the patient leaflet, and instructions on drug labels. Understanding and remembering treatment regimens are prerequisites for drug adherence [2]. However, approximately 40 to 80 per cent of the information during patient-physician encounters is forgotten or remembered inaccurately [2–4] and information in patient leaflets is often considered as too complex [5–7]. Therefore, patients would benefit from drug instructions that are easily read, understood, and remembered during their drug therapy. In this context, comprehensible drug label instructions (DLIs) may contribute to drug adherence. As drug labels are physically attached to each unit dispensed to patients, it is likely to be the last information source patients read before taking their drugs [8]. Therefore, DLIs should serve as an independent, comprehensible information source supporting patients' correct drug use which in turn facilitates their drug therapy [9].

The information on drug labels consists of the name of the patient, the name of the pharmacy, the name of the drug as well as the strength and the amount of the active substance in the drug [10]. Besides this information, drug labels consist of dosage instructions and auxiliary labels. Dosage instructions describe how patients should use the drugs, the intake frequency, and the number of units per intake (e.g., 'take two capsules twice daily'). Auxiliary labels consist of warnings (e.g., 'do not drink alcoholic beverages') and advices (e.g., 'take with food or milk'). The size and design of drug packages limit the amount of available space for DLIs, i.e., only the most essential instructions are presented in a concise way. This presentation might imply DLIs are easy to read and understand. However, research shows patients often misunderstand DLIs: patients misread label instructions, patients make errors when restating the instructions in their own words, and patients are unable to demonstrate a functional understanding (i.e., demonstrating when, how, and how many tablets they would take) [11–14].

Research shows patients' health literacy skills play a vital role in understanding DLIs and correct drug use [9,12,15–18]. The construct of health literacy deals with literacy skills in the context of health care, whereas literacy can be defined as basic skills in reading, writing, and numeracy [19]. Historically, health literacy was first used to describe the relation of patients' literacy level and their adherence to therapeutic regimens [20]. Gradually, the construct evolved to patients' skills to obtain, understand, and use health information in order to enhance health, well-being, and active involvement in medical decision making [21]. Different levels of health literacy can be distinguished: (1) functional health literacy: basic skills in reading and writing to be able to function effectively in everyday life, (2) interactive health literacy: advanced literacy skills and social skills are used to extract health information from different forms of communication and apply this information to changing circumstances, and (3) critical health literacy: advanced cognitive skills and social skills are used to critically analyse and

apply health information to exert greater control over life events and situations [20]. In sum, this classification shows functional health literacy is part of the broader construct of health literacy. Moreover, the levels of health literacy progressively allow for greater patient autonomy and empowerment [20,21].

Many different instruments have been developed to assess patients' health literacy which vary in adopted method and purpose [22]. For example, the Rapid Estimate of Adult Literacy in Medicine (REALM), Test of Functional Health Literacy in Adults (TOFHLA), and the Newest Vital Sign (NVS) all focus on functional health literacy, but differ in adopted method. The REALM tests the ability to read and pronounce 66 medical terms [23] whereas the TOFHLA consists of a 50-item reading comprehension and 17-item numerical ability test [24]. The NVS, on the other hand, consists of a reading comprehension and numerical ability test in which patients answer 6 questions about the information in a nutritional label for ice cream [25]. Other instruments, such as the Functional Communicative Critical Health Literacy (FCCHL) and the Set of Brief Screening Questions (SBSQ) assess patients' perceived health literacy skills, but differ in purpose and adopted method. For example, the FCCHL focuses on functional, interactive, and critical health literacy skills. It consists of 14 statements using 4-point Likert scales as response options [19]. The SBSQ only focuses on functional health literacy skills and consists of 3 statements using a 5-point Likert scale as response options [23] In sum, most instruments focus on functional health literacy skills, but the variation between the instruments in adopted method and purpose makes it hard to compare the results from health literacy interventions.

Patients with limited health literacy skills often experience difficulties in reading, understanding, and applying DLIs compared to patients with adequate health literacy [12,14,26]. Therefore, it is important to take health literacy into account when evaluating the comprehensibility of DLI's (cf., [27]). Especially as research shows 46 per cent of the Europeans have limited health literacy skills, varying from 29 per cent in the Netherlands to 62 per cent in Bulgaria [28]. Also, 14 per cent of the American population have limited health literacy skills [29]. As the processing of textual instructions poses a challenge for patients with limited health literacy, much research has been conducted on how visual elements, such as icons, pictograms, and graphics, can be used to improve the comprehensibility of DLIs [7,8,30]. However, mixed results have been found regarding their effectiveness. Wolf et al. [13] found simplified textual DLI with icons were more likely to be interpreted correctly by patients with marginal and limited health literacy compared to simplified textual DLIs alone. However, other studies report that icons on drug labels were frequently misunderstood by patients with limited health literacy [7,8,30]. As far as we know, in most countries the DLI attached on medication packages does not contain pictograms or graphics. Therefore, it is relevant to know how the textual instructions on drug labels are comprehended.

What is yet unknown, is how specific textual elements in DLIs (e.g., the presentation of numbers, or use of medical jargon) and patients' health literacy skills are related to (improved) comprehension of DLIs. While Samaranayake et al. [31] conducted a narrative review on the impact of patient-related factors, such as age and health literacy, and drug label-related factors, such as the use of icons and the format of the label, on the comprehensibility of DLIs, they did not provide an overview of specific textual elements associated with better comprehension of DLIs. Knowledge of which textual elements facilitates patients' comprehension of DLIs, especially those with limited health literacy, will provide concrete directions for health professionals, such as physicians and pharmacists, on how to optimize their drug prescriptions. In summary, the research question is: how are specific textual elements of DLI's and patients' health literacy related to the correct interpretation of DLIs? To answer this research question a systematic review was conducted of studies investigating the relation between the presence of specific textual elements in DLIs and patients' health literacy skills on the comprehensibility of DLIs.

## Method

### Search strategy

We conducted a literature search in PubMed, Embase, PsychINFO, and Smartcat to identify studies that examined the relation between textual elements and comprehensibility of DLIs. The search was limited to studies published in English or Dutch, up to April 2019. We searched these databases using various synonyms for 'drug labeling' and 'drug label comprehension'. Reference lists of relevant studies were manually checked, and relevant studies were included. The detailed search strategies are presented in Table 1.

### Study selection

Two reviewers (EM and HB) extracted data from the identified records, with one reviewer (EM) extracting data, and the other reviewer (HB) checking the information for accuracy. Duplicate publications listed in multiple databases were removed. Publications were eligible for inclusion if they met the following criteria: (1) studies which dealt with the comprehensibility of DLIs, and (2) studies which addressed textual elements of DLIs. Hence, articles focussing solely on the comprehensibility of DLIs in relation to icons, font, format, the way of printing, or other graphical elements were excluded. Also, articles about the comprehensibility of other health information sources than DLIs were excluded (reported as 'other information source' in Fig 1). Furthermore, only actual studies were included, which implies editorials, letters to the editor, or reviews were excluded (reported as 'no study' in Fig 1). Two reviewers (EM and HB) assessed whether studies had to be excluded based on titles and abstracts. To validate the assessment, twenty per cent of the identified titles and abstracts were independently annotated by the two reviewers (EM and HB) after which Cohen's kappas were calculated. The results showed the inter-rater reliability was good ($\kappa = 0.78$) [32].

After excluding studies based on titles and abstracts, both reviewers (EM and HB) read the full texts and assessed whether the study should be excluded, based on the eligibility criteria. Both reviewers had to agree on the eliminated studies. Differences in assessment were resolved by discussion or with assistance from a third reviewer (CVH or SB). Multiple publications reporting different analyses of data collected in the same study were counted as a single study with two separate research questions (see also data extraction).

### Data extraction

Data from all included articles were extracted using a data extraction sheet in Excel. The data extraction was reviewed by a second author (HB) for 33 per cent of the studies. As

**Table 1. Search strategy in electronic databases.**

| Database | Search components | Search |
|---|---|---|
| PubMed | Comprehension AND drug labeling OR prescription label | ((comprehension[MeSH Terms]) AND drug labeling[MeSH Terms]) OR prescription label[Title] |
| PubMed | Comprehension OR misunderstanding AND prescription label OR medication label | "Comprehension" (Mesh) OR misunderstanding (tiab)) AND (prescription label* (tiab) OR instruction label* (tiab) OR medication label* (tiab) Field: Title |
| SmartCat | Comprehension OR misunderstanding AND medication label OR prescription label | su: comprehension OR su:misunderstanding) AND (kw:medication label* OR kw: prescription label* |
| PsychINFO | Comprehension OR misunderstanding AND prescription label OR medication label OR instruction label | comprehension OR misunderstanding) AND (prescription label* OR medication label* OR instruction label* |
| Embase | Comprehension OR misunderstanding AND prescription AND label OR medication AND label OR instruction AND label | ('comprehension'/exp OR misunderstanding:ti,ab) AND (prescription:ti,ab AND label*:ti,ab OR (medication:ti,ab AND label*:ti,ab) OR (instruction:ti,ab AND label*: ti,ab)) |

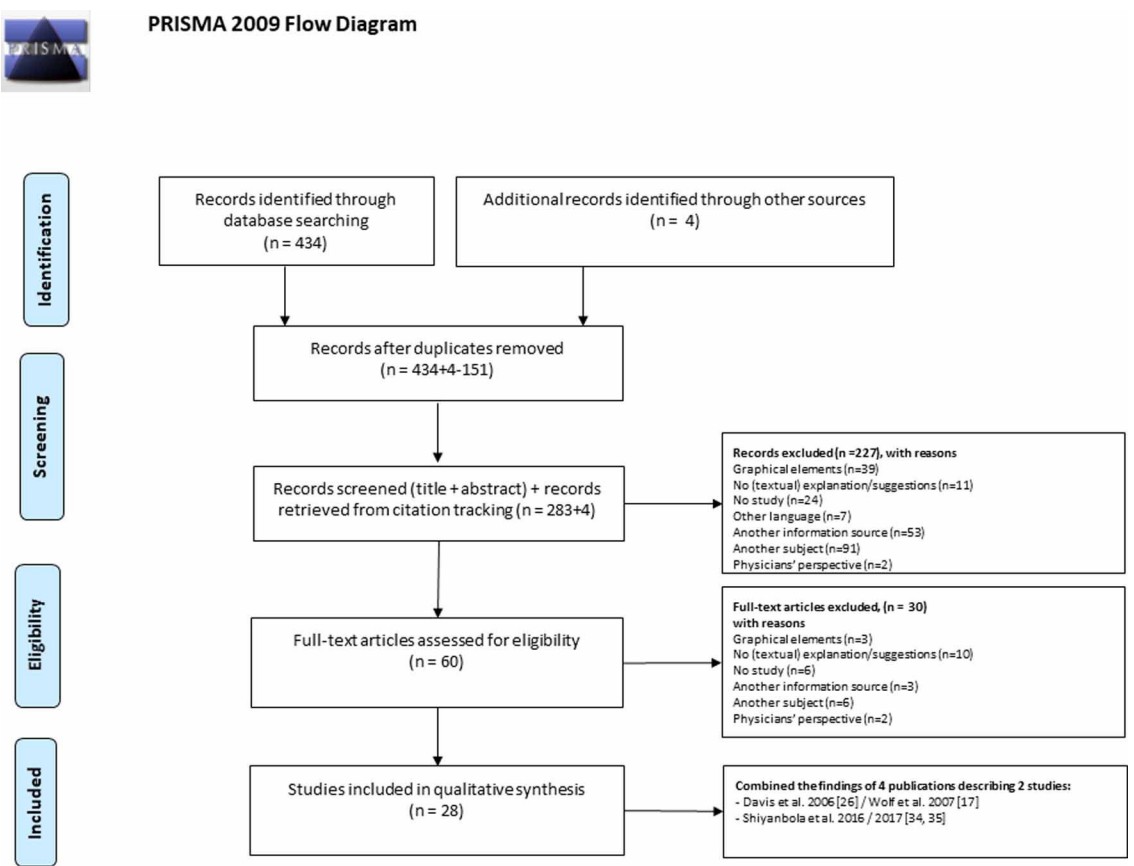

**Fig 1. Flow chart of the selection of studies about DLIs included in the review.**

there were no major differences between the reviewers, the remaining studies were only extracted by the first author (EM). Data extraction included: study type and setting, population, methods, primary outcome, textual elements, and conclusions. If the study compared an intervention and a control group, the intervention and the primary outcome was expressed as an effect of the intervention compared to the control group. If present, quantitative data were extracted concerning the impact of optimized DLIs compared to standard DLIs and differences between patients with limited health literacy and adequate health literacy on the primary outcome. Instruments measuring participants' health literacy were extracted to explore health literacy as explaining factor for differences in DLI comprehension. Missing information was scored as 'not reported'.

## Quality assessment of included studies

Data extraction was based on the Preferred Reporting Items for Systematic and Meta-Analysis (PRISMA) guidelines. This review included observational, cross-sectional, cohort, experimental, qualitative studies, and a randomized controlled trial. We used the Joanna Briggs Institute Critical Appraisal Checklists, for the cross-sectional studies, the cohort study, the experimental study, the qualitative studies and the randomized controlled trial. The checklists included 8 to 13 questions for assessing the methodological quality of the included studies (see Table 2) [33]. Two authors (EM and HvB) assessed the methodological quality of the included studies independently and the inter-rater reliability was, after discussion, 100%.

**Table 2. Assessment of the methodological quality of the included publications using the JBI critical appraisal checklists.**

| Risk of Bias Tools | Reference | Q1 | Q2 | Q3 | Q4 | Q5 | Q6 | Q7 | Q8 | Q9 | Q10 | Q11 | Q12 | Q13 |
|---|---|---|---|---|---|---|---|---|---|---|---|---|---|---|
| JBI Critical Appraisal Checklist for Cross-sectional studies | Alburikan et al., 2018 [36] | Yes | Yes | Yes | Yes | No | NA | Yes | Yes | | | | | |
| | Davis et al., 2006 [26] | Yes | Yes | Yes | Yes | No | NA | Yes | Yes | | | | | |
| | Koster et al., 2014 [41] | Yes | Yes | Yes | Yes | Yes | Yes | Yes | Yes | | | | | |
| | Locke et al., 2014 [42] | Yes | Yes | Yes | Yes | No | NA | Yes | Yes | | | | | |
| | Muluneh et al., 2018 [43] | Yes | Yes | Yes | Yes | Yes | Unclear | Yes | Yes | | | | | |
| | Davis et al., 2006 [12] | Yes | Yes | Yes | Yes | Yes | Yes | Yes | Yes | | | | | |
| | Shiyanbola et al., 2016 [35] | Yes | Yes | Yes | Yes | No | NA | Yes | Yes | | | | | |
| | Shiyanbola et al., 2017 [34] | Yes | Yes | Yes | Yes | No | NA | Yes | Yes | | | | | |
| | Wolf et al., 2006 [14] | Yes | Yes | Yes | Yes | No | NA | Yes | Yes | | | | | |
| | Wolf et al., 2010 [13] | Yes | Yes | Yes | Yes | Yes | Yes | Yes | Yes | | | | | |
| | You et al., 2011 [14] | Yes | Yes | Yes | Yes | Yes | Yes | Yes | Yes | | | | | |
| | Bailey et al., 2009 [11] | Yes | Yes | Yes | Yes | Yes | Yes | Yes | Yes | | | | | |
| | Beckman et al., 2005 [6] | Unclear | Unclear | Unclear | Unclear | No | NA | Yes | Yes | | | | | |
| | Comer et al., 2011 [49] | Yes | Yes | Yes | Yes | No | NA | Yes | Yes | | | | | |
| | Davis et al., 2009 [9] | Yes | Yes | Yes | Yes | No | NA | Yes | Yes | | | | | |
| | Holt et al., 1992 [38] | No | No | No | No | No | NA | Yes | Unclear | | | | | |
| | Kendrick & Bayne, 1982 [40] | No | No | Yes | Yes | No | NA | Yes | Unclear | | | | | |
| | Lo et al., 2006 [18] | Yes | Yes | Yes | Yes | No | NA | Yes | Yes | | | | | |
| | Mbuagbaw & Ndongmanji, 2012 [50] | Yes | Yes | Yes | Yes | Yes | Yes | Yes | Yes | | | | | |
| | McCarthy et al., 2013 [46] | Yes | Yes | Yes | Yes | Yes | Yes | Yes | Yes | | | | | |
| | Sahm et al., 2012 [37] | Yes | Yes | Yes | Yes | No | NA | Yes | Yes | | | | | |
| | Wallace et al., 2012 [39] | Yes | Yes | Yes | Yes | No | NA | Yes | Yes | | | | | |
| | Wolf et al., 2007 [17] | Yes | Yes | Yes | Yes | No | NA | Yes | Yes | | | | | |
| | Wolf et al., 2011 [47] | Yes | Yes | Yes | Yes | Yes | Yes | Yes | Yes | | | | | |
| | Hanchak et al., 1996 [53] | Yes | Yes | Yes | Yes | No | NA | Yes | Yes | | | | | |
| JBI Critical Appraisal Checklist for Experimental study | Bailey et al., 2012 [5] | Yes | No | Yes | Yes | NA | Yes | Yes | Yes | Yes | | | | |
| JBI Critial Appraisal Checklist for Qualitative studies | Webb et al., 2008 [8] | Yes | Yes | Yes | Yes | Yes | No | No | Yes | Yes | Yes | | | |
| | Bailey et al., 2014 [45] | Yes | Yes | Yes | Yes | Yes | No | Yes | Yes | Yes | Yes | | | |
| JBI Critial Appraisal Checklist for Randomized Controlled Trials | Wolf et al., 2016 [51] | Yes | No | No | NA | No | No | Yes | Yes | Yes | Yes | Yes | Yes | Yes |
| | McManus et al., 2018 [48] | Yes | Yes | Yes | No | No | No | Yes | Yes | Yes | Yes | Yes | Yes | Yes |

## Data analysis

Data were synthesized by narrative and tabular methods. The included studies differed substantially regarding patient population and measurements methods. The study characteristics (primary outcome, study design and sample, patient characteristics, health literacy skills, and textual elements) of the publications were classified (Table 3). The studies were grouped by DLI type. Three types were distinguished: dosage instructions, auxiliary labels, and drug labels with combined instructions. The latter category was used when a study investigated a combination of auxiliary labels and dosage instructions, or when the DLI type was not specified.

**Table 3. Characteristics of included studies (author(s), primary outcome, study design and sample, and health literacy) and recommendations on composing comprehensible DLIs, stratified to type of instruction (drug label, auxiliary label instruction, or dosage instruction) and type of textual element (plain language, native language, presentation of numbers, specificity of dosage instructions, number of messages per instruction line, and abbreviations).**

| | Author(s), publication year | Primary outcome | Study design and sample | Patients and Health Literacy (HL) measure | Type of instructions | Findings on textual elements (TE) and health literacy (HL), if reported | Textual elements | Recommendations concerning textual elements |
|---|---|---|---|---|---|---|---|---|
| | | | | DRUG LABELS WITH COMBINED INSTRUCTIONS | | | | |
| 1. | Alburikan et al., 2018 [36] | Prescription label comprehension was measured by asking patients 'How would you take this medicine?' for commonly prescribed drugs. | A multicentre, cross-sectional observational study based on semi-structured interviews. | 511 Saudi patients. HL: Arabic Single Item Literacy Screener. | Arabic drug labels. | TE: The hybridization of the label language between Arabic and English lead to misunderstanding. All of the label aspects were written in Arabic except the numbers, which were written in English. TE: Other causes for misunderstanding were complexity of the DLI and implicit dosage intervals. HL: Patients with limited HL skills were associated with a high percentage of misunderstanding DLIs in comparison to patients with good HL skills (59.5% versus 36.0%). | - Instruction in native language - Presentation of numbers - Specificity of dosage instructions | - Write instructions (both text and numbers) in patients' native language. - Instructions containing fewer numbers enhance comprehensibility (e.g., *'Take 1 tablet by mouth 1 time a week'* vs. *'Take 5 ml every 8 hours for 7 days'*). - Provide instructions with explicit dosage intervals. |
| 2. | Bailey et al., 2012 [5] | Prescription label comprehension was measured by asking participants 'Using this dosing tray, please show me when you would take this drug over the course of one full day.' Prescription label regimen dosing and consolidation was measured by handing participants five medication bottles. Participants used the dosing tray to demonstrate how and when they would take all five drugs on a typical day. | Randomized experimental evaluation. | 202 Low English Proficiency (LEP) adult speaking 5 non-English languages. HL: not reported. | ConcordantRx instructions (i.e., multilingual instructions which incorporated health literacy 'best practices') vs. standard instructions. | TE: Participants receiving the ConcordantRx instructions demonstrated significantly greater comprehension, regimen dosing and regimen consolidation compared to those receiving standard instructions. HL: - | - Specificity of dosage instructions - Plain language - Presentation of numbers - Instruction in native language | - Specify dosage intervals to four distinct time periods (i.e., *'morning, noon, evening, bedtime'*). - Use simple terms (e.g. *'pills'* vs. *'tablets'*). - Use numbers instead of letters (i.e., *'2'* vs. *'two'*). - Translate instructions to patients' native language. |

*(Continued)*

Table 3. (Continued)

| | Author(s), publication year | Primary outcome | Study design and sample | Patients and Health Literacy (HL) measure | Type of instructions | Findings on textual elements (TE) and health literacy (HL), if reported | Textual elements | Recommendations concerning textual elements |
|---|---|---|---|---|---|---|---|---|
| 3a**. | Davis et al., 2006 [26] | Prescription label comprehension was measured by asking patients 'How would you take this medicine?' Functional prescription label comprehension was measured by asking patients 'Show me how many pills you would take of this medicine in one day'. | Cross-sectional study using in-person, structured cognitive interviews. | 395 adults. HL: REALM. | Container label instructions and warning labels. | TE: The majority (51.8%) of incorrect patient responses reflected an error in dosage (i.e., 'tablespoon' vs. 'teaspoon') and 28.2% stated the wrong dose frequency (i.e., 'one tablet each day for seven days' vs. 'Take one tablet by mouth twice daily for seven days'). HL: Patients reading at or below the 6th grade level (limited literacy) were less able to understand all 5 label instructions. Prevalence of misunderstanding of one or more DLI was for patients with adequate, marginal, and limited literacy was 37.7%, 51.3%, and 62.7%, respectively. | - Presentation of numbers<br>- Specificity of dosage instructions<br>- Dose measurements | - Instructions containing fewer numbers enhance comprehensibility (i.e., 'Take one tablet by mouth once each day' vs. 'Take one tablet by mouth twice daily for seven days').<br>- Explicate dosage instructions with daypart- and hour of intake (i.e., 'Take one tablet in the morning and one at 5 p.m.').<br>- The formulation of dose measurements should be evaluated. |
| 3b**. | Wolf et al., 2007 [17] | See study 3a | See study 3a. | See study 3a. | See study 3a. | TE: Patients were better able to interpret more explicit dose frequencies, such as 'Take one tablet in the morning and one at 5 p.m' (90%), compared to 'Take two tablets by mouth twice daily' (83%), and 'Take one teaspoonful by mouth three times daily' (73%). TE: Patients found simpler dosing regimens easier to understand while more complex regimens resulted in more errors in their interpretation. HL: The prevalence of misunderstanding among patients with adequate, marginal and limited literacy was 38%, 51% and 63% respectively. HL: Patients with low literacy were less able to state the correct number of pills taken daily compared to those with marginal and adequate literacy (35% versus 63% versus 80) | - Plain language<br>- Specificity of dosage instructions | - Avoid repetitiveness in numbers between dosage ('two') and frequency ('twice').<br>- Words as 'antibiotic', 'orally', 'teaspoonful', 'medication', 'prescription' and 'dose' were difficult to recognize and pronounce by patients reading at 6th grade level or below.<br>- Avoid unclarified medical jargon ('antibiotic') or awkward terms ('twice').<br>- The amount of content to be retained (dosage, frequency and/or duration of intake) makes instructions complex.<br>- Explicit dose frequencies were easier to interpret than implicit, vague dose frequencies ('Take one tablet in the morning and one at 5 p.m' vs. 'Take two tablets by mouth twice daily'). |

(Continued)

**Table 3.** (Continued)

| | Author(s), publication year | Primary outcome | Study design and sample | Patients and Health Literacy (HL) measure | Type of instructions | Findings on textual elements (TE) and health literacy (HL), if reported | Textual elements | Recommendations concerning textual elements |
|---|---|---|---|---|---|---|---|---|
| 4. | Koster et al., 2014 [41] | Participants completed a survey containing multiple-choice questions about five frequently used standard drug label instructions (i.e., combination of over the counter and prescribed drugs). | Cross-sectional study. Individually completed surveys. | 691 First-generation immigrants from the Antilles, Iran and Turkey. Reference group: 153 Dutch first-year pharmacy students. HL: not reported. | Five DLIs. | TE: Only two out of five DLI's were interpreted correctly by the majority of all respondent groups. The instruction *'Take with water, not with milk'* was correctly interpreted by less than 25% of the participants. The warning *'Avoid sunlight exposure'* was often misinterpreted (<8% correct). The instruction *'Do NOT drink grapefruit juice along with this drug'* was correctly interpreted by approximately half of the participants HL: - | - Plain language<br>- Instruction in native language | - Reformulate *'Take with water, not with milk'* by mentioning dairy products should be avoided.<br>- Reformulate the instruction *'Do NOT drink grapefruit juice along with this drug'* by *'Do not use grapefruit(juice) while taking this medicine'.*<br>- Instructions should be given preferably in the patients' native language. |
| 5. | Locke et al., (2014) [42] | Comprehension of new and existing prescription auxiliary labels was measured by asking patients 'What do you think this auxiliary label is saying?' | Cross-sectional study. Semi-structured interviews. | 120 Adults from minority populations (all ethnicities other than non-Hispanic white) who were currently taking or had previously taken a prescription medication and could understand English. HL: Revised REALM (REALM-R). | Four existing prescription auxiliary labels. New labels were designed based on previous research. | TE: All existing prescription auxiliary labels yielded less than 50% correct interpretation except for *'Take with food'* and *'Do not chew or crush'.* The newly designed labels were better understood compared to existing labels. HL: 56.7% had a REALM-R score below 7th grade. Higher REALM-R score was associated with better interpretation of DLIs. | - Number of messages per instruction line<br>- Plain language | - Newly designed DLIs were single item instructions (e.g., *'Swallow this medication whole'*).<br>- Newly designed DLIs were written in plain language (e.g., *'Take this medication with a full glass of water'*). |
| 6. | Muluneh et al., (2018) [43] | Cancer patients' use of oral chemotherapies and comprehension of labelling directions was investigated using a 30-question survey. | Survey. | 93 Patients taking oral chemotherapies, adult patients (>18 years old), diagnosed with breast cancer, renal cell carcinoma (RRC), chronic myeloid leukaemia (CML) and colorectal cancer (CRC). HL: not reported. | Drug labels of oral chemotherapy. | TE: 15% of the patients had difficulty understanding label directions. Patients had recommendations to help them understand textual aspects of the medication label: avoiding abbreviations (23%) and easier directions (21%). HL: - | - Abbreviations<br>- Plain language | - Abbreviations should be avoided.<br>- Uses easier label directions. However, it is not mentioned what easy directions are. |
| AUXILIARY LABEL INSTRUCTIONS |
| 7. | Davis et al., 2006 [12] | Comprehension of commonly used prescription medication warnings was measured by asking patients 'What does this label mean to you?' | In-person structured interviews. | 251 Patients receiving primary care at a hospital clinic. HL: REALM. | Warning labels. | TE: The simplest label (*'Take with food'*) was interpreted correctly by 84% of the patients while a label with multi-step instructions like *'Do not take dairy products, antacids or iron preparations within 1 hour of this medication'* was interpreted correctly by 7.6% of the patients. HL: Patients with limited HL were at greater risk to misinterpret DLIs. | - Number of messages per instruction line | - Use single action instructions (e.g., *'Take with food'*. |

*(Continued)*

**Table 3.** (Continued)

| | Author(s), publication year | Primary outcome | Study design and sample | Patients and Health Literacy (HL) measure | Type of instructions | Findings on textual elements (TE) and health literacy (HL), if reported | Textual elements | Recommendations concerning textual elements |
|---|---|---|---|---|---|---|---|---|
| 8a**. | Shiyanbola et al., 2016 [35] | Patients' perspectives on the words (content) of prescription warning labels were examined. | Semi-structured face-to-face interviews. | 21 patients. HL: NVS. | Warning labels. | TE: Patients wanted the warning instructions to be more specific, especially the time frame needed to adhere to the instruction. TE: For the instruction '*Take with a full glass of water*' patients preferred more information about the meaning of '*a full glass*'. TE: Patient preferred the inclusion of the word '*warning*' on the PWL to create alertness. HL: The mean HL score was 2.4. Prevalence of patients with limited, marginal and adequate HL was 19%, 14% and 43% respectively. | - Plain language | - Specify the time frame needed to adhere to the instruction. - Specify '*a full glass of water*'. - Include the word '*warning*' on the label and the reason for the warning. |
| 8b**. | Shiyanbola et al., 2017 [34] | Pharmacists' perspectives on the words (content) of prescription warning labels were examined. | See study 8a. | 8 Pharmacists. | See study 8a. | TE: Pharmacists wanted the instructions to be more specific, especially the time frame needed to adhere to the instruction. For the instruction '*Take with a full glass of water*' they preferred more information about the meaning of '*a full glass*'. | - Plain language | - Specify the time frame needed to adhere to the instruction. - Specify '*a full glass of water*'. |
| 9. | Webb et al., 2008 [8] | Comprehension of prescription warning labels was investigated by showing participants a prescription pill bottle container with a prototype of a warning label asking them to interpret the label. Participants were then tested on their comprehension of 10 common warning labels by matching warning messages with the corresponding icons. Participants took part in a discussion group to solicit feedback around improving existing language and content, and revising icons of ten of the most commonly used warning labels. Patients were shown warning message in text form only and asked 'If this message was on your prescription pill bottle, how would you take the medicine?' | Structured cognitive interviews followed by discussion groups that solicited feedback for revising text and icons. | 85 adults. HL: REALM. | Warning labels. | TE: Most text messages were confusing and used language that was too difficult. HL: 56% of patients had limited HL. | - Plain language | Language should be shortened and made clear and simple. Words like '*external*', '*prolonged or excessive exposure*' and '*non-prescription*' should be avoided. '*External*' should be written as '*on your skin*', '*Prolonged or excessive exposure*' as '*limit your time in the sun*' and '*non-prescription*' as '*over-the-counter*'. |

*(Continued)*

**Table 3.** (Continued)

| | Author(s), publication year | Primary outcome | Study design and sample | Patients and Health Literacy (HL) measure | Type of instructions | Findings on textual elements (TE) and health literacy (HL), if reported | Textual elements | Recommendations concerning textual elements |
|---|---|---|---|---|---|---|---|---|
| 10. | Wolf et al., 2006 [14] | Comprehension was measured by asking patients to interpret and comment on eight commonly used warning labels on prescribed drugs. | Structured interviews. | 74 Patients reading at 6th -grade level or below. HL: REALM. | Warning labels. | TE: Rates of correct interpretation of the eight prescription warning labels ranged from 0% to 78.7%. None of the patients were able to correctly interpret 'Do not take dairy products, antacids, or iron preparations within one hour of this medication'. The causes for misunderstanding were attributed to word choice, message length, and number of steps for action. HL: patients 3rd-grade or below: 38%. Patients 4th-6th grade: 62%. Patients with low literacy skills demonstrated a lower rate of correct interpretation of the eight most commonly used prescription warning labels than did those with higher literacy. | - Number of messages per instruction line - Plain language | - Avoid multiple-step instructions (e.g., 'Refrigerate, shake well, discard after . . .') - Avoid unnecessary complex (e.g., 'You should avoid prolonged or excessive exposure to direct and/or artificial sunlight while taking this medication') or vague formulations (e.g., 'medication should be taken with plenty of water'). - Refrain from professional jargon (e.g., 'iron preparations', 'dairy products', 'antacids'). |
| 11. | Wolf et al., 2010 [13] | Patients' interpretations of the nine (prescription and over-the-counter) drug warnings placed on container vials was investigated by asking 'In your own words, what do these labels mean to you?' | Semi-structured interviews. | 500 adult patients. HL: REALM. | Warning labels: (1) current standard drug warning labels, or (2) drug warnings with text rewritten in plain language, or (3) plain language and icons developed with patient feedback. | TE: Auxiliary warning labels with explicit, easy-to-read messages significantly improved rate of attendance and comprehension among patients. Prescription drug warning labels with simplified text and simplified text + icons were significantly more likely to be correctly interpreted compared with standard labels. HL: 20.1% were reading below 7th grade level (= limited literacy). Limited literacy was an independent predictor of misinterpretation. | - Plain language | Use plain language in DLIs, such as 'Limit your time in the sun' |
| 12. | You et al., 2011 [44] | Patients' interpretation of prescription warning labels was examined by asking 'In your own words, what does this mean to you?' | Structured cognitive interview. | 132 Pregnant and/or breast-feeding females at reproductive age. HL: REALM. | Warning labels: (1) current teratogen warning, (2) label with simplified text and (3) label with simplified text and icons. | TE: Comprehension of enhanced text + icon label (94%) was significantly higher than the standard (76%) and enhanced text-only (79%) labels. HL: 18% low HL, 39% marginal HL and 42% adequate HL. Interactions on HL were not found to be significant. | - Plain language | Use explicit, easy-to-read messages in plain language. However, authors do not explicate what plain language and easy-to-read messages are. |

DOSAGEINSTRUCTIONS

(*Continued*)

Table 3. (Continued)

| # | Author(s), publication year | Primary outcome | Study design and sample | Patients and Health Literacy (HL) measure | Type of instructions | Findings on textual elements (TE) and health literacy (HL), if reported | Textual elements | Recommendations concerning textual elements |
|---|---|---|---|---|---|---|---|---|
| 13. | Bailey et al., 2009 [11] | Participants' comprehension of dosage instructions of a liquid drug commonly prescribed for children was investigated. To assess participants' understanding of the prescription labels they were asked 'How would you give this medicine?' | Qualitative structured interviews. | 373 adults. HL: REALM. | Dosage instructions of liquid drugs. | TE: One in three participants misunderstood dosage instructions. Common causes for misunderstanding included problems with dosage measurement (28%; i.e., 'tablespoon' vs. 'teaspoon') and frequency of use (33%; i.e., 'every 3 hours' vs. 'every 6–8 hours'). HL: 43.2% of patients with limited HL misunderstood DLIs. Inadequate and marginal HL were independent predictors of misunderstanding DLIs | - Dose measurements - Specificity of dosage instructions | No explicit recommendation mentioned, authors mention that studies are underway to evaluate the formulation of dose measurement for liquid medication. Separate dose from interval and provide the explicit frequency of the drug (e.g., 'Take 1 (unit) at morning, take 1 (unit) noon, and take 1 (unit) at bedtime' vs. 'Take one teaspoon full by mouth three times daily'). |
| 14. | Bailey et al., 2014 [45] | During discussion groups, participants were shown a series of 'standard' and 'improved' prescribed drug instructions. Participants were asked to consider how the terminology and phrasing of instructions could be improved to promote comprehension. | Four iterative sessions of discussion groups. | 40 English speaking adults. HL: REALM. | Dosage instructions: standard instructions (e.g., 'Take one inhalation twice a day') vs. improved instructions (e.g. 'Take 1 puff in the morning and 1 puff in the evening every day') which incorporated the UMS and health literacy best practices: - Chunking/grouping information - Easy-to-understand terms - Explicit dosage information - Specific time periods - Sentence format - Numbers in numeric form | TE: Participants suggested using clear, concise wording and phrasing whenever possible. Words, such as 'subcutaneously' and 'inhalation', were viewed as unnecessarily difficult, by many participants. TE: Participants preferred 'half' over '0.5' and '½'. TE: Participants agreed on the need detailed instructions (i.e., 'when you are short of breath' vs. 'as needed'; 'until all pills have been taken' vs. 'until the bottle is empty'). HL: 33% of the patients had limited HL skills. | - Dose measurements - Presentation of numbers - Plain language | - Authors recommend additional research on the comprehensibility of measurement formulations ('teaspoon' vs. 'teaspoonful'). - Use a numeric instead of an alphanumeric presentation for numbers, except for fractions. - Avoid the term 'as needed'. - The duration and maximum dose per day should be specified on the label for short-term drugs. - Avoid the word 'maximum'. Use 'Do not take more than … per day'. - Avoid unnecessarily difficult words, such as 'subcutaneously' and 'inhalation'. Use 'under your skin' and 'puff'. |

(Continued)

**Table 3.** (Continued)

| | Author(s), publication year | Primary outcome | Study design and sample | Patients and Health Literacy (HL) measure | Type of instructions | Findings on textual elements (TE) and health literacy (HL), if reported | Textual elements | Recommendations concerning textual elements |
|---|---|---|---|---|---|---|---|---|
| 15. | **Beckman et al., 2005** [6] | Comprehension of DLIs was measured using three tests: 1) Participants were given a box of aspirin. They were asked to read the instructions and answer the question 'What is the maximum number of times you may take this aspirin during one day? 2) Participants were given a box of penicillin with the instruction '2 tablets morning and evening'. They were asked how many days the pills would last. 3) Participants were given a receipt from the pharmacy with the sum of two items billed for a total of 64 crowns. They were asked to calculate how much change they would receive if they paid with a 100-crown note. | Performance of three cognitive tests through direct interviews. | 492 Elderlies in Sweden, aged 77 or older. HL: not reported. | Dosage instructions. | TE: For the first test with the aspirin container (i.e., understanding instructions), 30.7% answered incorrectly. The second test with the penicillin instruction (i.e., calculating number of days) 47.4% answered incorrectly. For the third test in which participants had to calculate their change at the pharmacy, 20% answered incorrectly. HL: - | Specificity of dosage instructions | The instruction '2 tablets morning and evening' should be rephrased as '2 tablets in the morning and 2 tablets in the evening'. |
| 16. | **Comer et al., 2011** [49] | The variability in patients' understanding of quantitative statements from prescription orders was investigated. Patients received a scenario and were asked to use a proved tube of cream with the instruction 'Apply a small amount to the area' and squeeze out what they considered as a small amount. | Interviews. | 100 patients. HL: not reported. | Dosage instruction 'Apply a small amount to the area' | TE: Patients showed variability in the interpretation of a small amount of topical product cream. The mean weight of a small amount was between 36 and 50 grams. HL: - | Dose measurements | Be specific in what is meant by phrases like 'a small amount'. However, the authors do not provide alternative formulations. |
| 17. | **Davis et al., 2009** [9] | Comprehension of prescription DLIs was measured by showing patients ten prescription bottles one at a time and asking 'How would you take this medicine?' | Cross-sectional study using in-person, structured interviews. | 359 adults. HL: REALM. | Dosage instructions. | TE: Patients were significantly more likely to understand instructions with explicit time periods compared to instructions stating times per day or hourly intervals. Four out of five patients in this study misinterpreted one or more of the ten common DLIs. HL: 55 patients had low HL, 109 patients had marginal HL and 195 patients had adequate HL. Low and marginal literacy were significant independent predictors of misinterpreting instructions. | Specificity of dosage instructions | Avoid frequency in hourly intervals or the number of times of day in dosage instructions (e.g., 'Take 1 pill by mouth every 12 hours with a meal'; 'Take two tablets by mouth twice daily'). Instead use time periods in dosage instructions (e.g., 'Take 2 pills in the morning and 2 pills in the evening'; 'Take 1 pill by mouth every day') |

*(Continued)*

**Table 3.** (Continued)

| | Author(s), publication year | Primary outcome | Study design and sample | Patients and Health Literacy (HL) measure | Type of instructions | Findings on textual elements (TE) and health literacy (HL), if reported | Textual elements | Recommendations concerning textual elements |
|---|---|---|---|---|---|---|---|---|
| 18. | Hanchak et al., 1996 [53] | Comprehension of prescribed drug dosage instructions was investigated by asking patients 'How many times a day do you understand that your medication is to be taken?' | Prospective cohort study. | 500 patients. HL: not reported. | Dosage instructions. | TE: Dosage instructions specifying hourly intervals were less understood than dosage instructions specifying daily frequency. HL: - | Specificity of dosage instructions | Use dosage instructions in which daily frequency is specified instead of dosage instruction in which hourly intervals are specified |
| 19. | Holt et al., 1992 [38] | Consumers' interpretations of dosage instructions was investigated by asking consumers how they would take drugs. The DLIs were common and could occur on over-the-counter and prescribed drugs. | Survey. | 321 participants. HL: not reported. | Six dosage instructions: Take 1 tablet 3 times daily. Take 1 tablet every 8 hours. Take 1 tablet daily. Take 1 tablet twice daily. Take 1 tablet every 12 hours. Take 2 tablets daily. | TE: Dosage instructions elicited a higher percentage of correct responses if they specified the number of hours between doses (i.e., 'Take 1 tablet every 8 hours') compared to dosage instructions which did not specify the number of hours between doses (i.e., Take 1 tablet 3 times daily'). The use of specific hourly intervals is superior to more vague instructions. The use of daily terms is more frequently associated with correct comprehension than meal terms. HL: - | Specificity of dosage instructions | Explain specifically how medications are to be taken by using specific hourly intervals (morning, noon, evening, bedtime). |
| 20. | Kendrick & Bayne, 1982 [40] | Comprehension was measured by asking patients how many of their prescribed drug they should take every day. Also, patients were asked to explain how many tablets a day they would take following the instruction: 'Take one tablet every 6 hours'. | Interviews with questions on: 1) where they kept their medicines; 2) how many pills they should be taking every day. | 40 Patients, 65 years or older of age. HL: not reported | Dosage instruction 'Take one tablet every 6 hours' | TE: In 29% of the cases, participants' understanding differed from what was written on the drug label. For the instruction 'Take one tablet every 6 hours', only 22% of 37 participants answered that they would take four pills in a day HL: - | Specificity of dosage instructions | Oral instructions should be reinforced with explicit written instructions. |
| 21. | Lo et al., 2006 [18] | Comprehension of a DLI of a prescription for ferrous sulfate was investigated. Participants answered the following questions: 1) Please use the medicine dropper to show me how much medicine you should give. 2) How many times a day should you give this medicine? 3) If you give the first dose now, when should you give the next dose of medicine? | An anonymous cross-sectional survey. | 326 English-speaking parents who were 18 years or older, primary caretaker of a child of 5 years or younger. HL: A quantitative portion of the TOFHLA to indicate parental HL. | Dosage instruction 'Give 1 dropperful by mouth twice daily' | TE: 252 (77%) participants incorrectly dosed the medicine. The majority of errors occurred with demonstrating the amount of medicine to give (74%) and stating the time to give the next dose (67%). Participants did not understand the term 'dropperful'. HL: No numbers reported about the HL of patients. | - Dose measurements - Specificity of dosage instructions | - Avoid 'dropperful' to indicate dose measurements. - Specify the number of hours in between doses or the hour to give each dose. |

*(Continued)*

**Table 3.** (Continued)

| | Author(s), publication year | Study design and sample | Primary outcome | Patients and Health Literacy (HL) measure | Type of instructions | Findings on textual elements (TE) and health literacy (HL), if reported | Textual elements | Recommendations concerning textual elements |
|---|---|---|---|---|---|---|---|---|
| 22. | Mbuagbaw & Ndongmanji, 2012 [50] | Cross-sectional study; a pilot tested questionnaire. | Comprehension of frequently used prescriptions was measured by asking patients if they could tell how drugs should be taken. | 204 Outpatients in semi-urban Cameroon. HL: not reported. | Four different DLIs: fully written out, Latin abbreviations, symbols, and pictograms | Latin abbreviations were least understood (26.9%), DLIs with symbols were understood by 89.7% of participants and written-out instructions by 87.7% of participants. HL: - | Abbreviations | Written-out instructions are better understood. Avoid the use of Latin abbreviations. |
| 23. | McCarthy et al., 2013 [46] | In-person interviews. | Comprehension was measured by presenting participants a dosing tray that contained 24 slots representing each hour of a day. Participants were given a hypothetical prescription bottle and the scenario that they had been prescribed pain medication. Participants answered the following questions: 1) Imagine that it is 8am and you are having pain. Please show me how many pills of this medicine you would take at 8am by placing the beads into the box.' 2) If you were still in pain after taking this dose of medicine, when could you take this medicine again? 3) Show me at what time and how many pills of this medicine you would take for your next dose. | 87 adults. HL: REALM. | Standard 'as needed' instructions versus a patient-centred Take-Wait-Stop label. The Take-Wait-Stop label included explicit, deconstructed instructions along with simplified text, numeric characters instead of words, and 'carriage returns' to place each part of the instructions on separate lines. In addition, to convey the maximum daily dosage the word 'Stop' was used to replace the typical wording 'Do not exceed'. | TE: 31.8% of the participants who were shown the standard label demonstrated taking in excess of 6 pills in 24 hours compared with only 14.0% of participants who were shown the Take-Wait-Stop label. TE: Of the standard label group, 20.5% of the participants demonstrated dosing intervals of fewer than 4 hours compared with 23.3% of the participants the Take-Wait-Stop label group TE: Participants who were exposed to the standard label were 2.5 times more likely to exceed the recommended maximum daily dose. HL:72.4% had adequate literacy. Study was not powered to detect such differences, large extent had adequate HL. | Plain language, specificity of dosage instructions, presentation of numbers, single item instructions | Use Take-Wait-Stop instructions. Example: *'Take: 1 or 2 pills Wait: 4 hours before taking again Stop: Do not take more than 6 pills in 24 hours'* |
| 24. | Sahm et al., 2012 [37] | In-person interviews. | Correct interpretation of three prescribed dosage instructions was evaluated by asking participants the following questions: 1) In your own words, how would you take this medicine? 2) How many tablets would you take of this medicine in one day? | 94 participants. HL: REALM. | Different dosage instructions: 1) Standard prescription instructions written as times per day (usual care). 2) Patient centred instructions hat specify explicit timing with standard intervals (morning, noon, evening, bedtime) or with mealtime anchors. 3) Patient centred instructions with a graphic aid to visually depict dose and timing of the medication. | TE: PCI were more likely to be correctly interpreted than the standard instructions. HL: 30.9% with limited HL. Patients with limited health literacy were more likely to correctly interpret the patient centred instructions (91%) than the standard instructions (66%). | Specificity of dosage instructions | Specify explicit timing with standard intervals: morning, noon, evening, bedtime or mealtime-anchors. |

(*Continued*)

Table 3. (Continued)

| | Author(s), publication year | Primary outcome | Study design and sample | Patients and Health Literacy (HL) measure | Type of instructions | Findings on textual elements (TE) and health literacy (HL), if reported | Textual elements | Recommendations concerning textual elements |
|---|---|---|---|---|---|---|---|---|
| 25. | Wallace et al., 2012 [39] | Comprehension of dosage instructions was measured by presenting participants a bottle of over-the-counter medicine. Participants answered two questions: 1) Please describe how you would give this medicine over a 24-hour period (1 day). 2) Please show me how you would give one dose of this medicine. Participants could use three oral liquid measuring devices. | A structured interview. | 193 English-speaking women of childbearing age. HL: SBSQ-D. | Implicit dosage instruction: 'Shake liquid well and give (child's name) 6ml by mouth every 12 hours' Explicit dosage instruction: 'Shake liquid well and give (child's name) 6 ml by mouth at 7 am and 7 pm'. | TE: Approximately one third of participants (32.1%) were able to describe and measure a dose of the medication correctly. However, implicit versus explicit dosage intervals—did not result in improved patient comprehension of instructions. HL: 48.7% of the participants had inadequate HL. HL was associated with higher odds of correctly measuring a dose of the medication. The prevalence of understanding among patients with adequate, marginal and limited literacy was 82.8%, 74.1% and 55.6% respectively. | Specificity of dosage instructions | Specify the dosing interval |
| 26. | Wolf et al., 2011 [47] | Comprehension was measured by providing patients a hypothetical drug regimen, which consisted of 7 actual prescription drug pill bottles with mock-up labels, each with a retired drug name and different dosage instructions. Patients had to demonstrate when they would take the entire regimen by dosing fake pills contained with each prescription bottle at the times of day that they would take the drugs. | A structured interview. | 464 adult patients. HL: REALM. | Dosage instructions with implicit timing intervals (e.g., 'Take 1 tablet by mouth 3 times daily') and dosage instructions with explicit timing intervals (e.g., 'Take 1 tablet by mouth at bedtime') | TE: One-third of the participants (29.3%) dosed their medications 7 or more times per day, while only 14.9% organized the regimen into 4 or fewer times a day. TE: When the drugs had variable expressions of the same dose frequency (e.g., 'every 12 hours' vs. 'twice daily'), 79.0% of the participants did not consolidate the drugs. HL: Nearly half of the participants were identified as having either low (20.7%) or marginal (22.8%) health literacy skills. HL: Low health literacy was found to be the sole independent predictor of a greater number of times per day for dosing the 7-drug regimen. | Specificity of dosage instructions | Provide standard, explicit instructions. However, the authors do not provide examples of standard, explicit instructions. |

*(Continued)*

**Table 3.** (Continued)

| | Author(s), publication year | Study design and sample | Primary outcome | Patients and Health Literacy (HL) measure | Type of instructions | Findings on textual elements (TE) and health literacy (HL), if reported | Textual elements | Recommendations concerning textual elements |
|---|---|---|---|---|---|---|---|---|
| 27. | Wolf et al., 2016 [51] | Two-arm, multi-site patient-randomized pragmatic trial. Interviews. | Comprehension of a prescribed drug in a patient's regimen was measured at the baseline, 3 months, and 9 months by patient's ability to correctly report, for each medication: 1) How many pills taken per dose. 2) Times per day a medicine was to be taken, specifying the hour of each dose. 3) The total number of pills taken daily. Patients' adherence was measured at 3 and 9 months via: 1) Self-report of missed or incorrect doses in the prior 4 days using the Patient Medication Adherence Questionnaire. 2) Pill count (for diabetes and hypertensive medicines). | 845 English- or Spanish speaking patients, ≥30 years of age, diagnosed with type 2 diabetes and/ or hypertension and taking ≥2 oral medications. HL: REALM for English speaking patients and SAHLSA for LEP Spanish speaking patients. | Standard label instructions vs. patient-centred label instructions. The patient-centred label instructions incorporated evidence-based practices for format and content, including prioritized information, larger font size, and increased white space. Most notably, patient-centred instructions were conveyed with the UMS, which uses standard intervals for expressing when to take medicine (morning, noon, evening, bedtime). | TE: Patients receiving the patient-centred instructions demonstrated slightly better proper use of their drug regimens at first exposure and at 9 months. However, patient-centred instructions did not improve drug adherence. HL: 37.4% limited HL and 62.6% adequate HL. The patient-centred instructions were particularly better understood by patients with limited literacy. | Specificity of dosage instructions | Specify dosage instructions using standard time intervals (morning, noon, evening and bedtime). |
| 28. | McManus et al., 2018 [48] | A pilot randomised controlled trial. | Comprehension was measured by showing patients five prescription drugs. For each drug participants were asked the following questions: 1) How many tablets would you take at any one time? 2) How many tablets would you take in a day? 3) Are there any precautions you would take while taking this medicine? In addition, participants were asked to dose out the five drugs into a 24h dosette box. | 76 adult in-patients, receiving oral medicines, who spoke English fluently. HL: NVS and validated HL screening questions. | Standard dosage instructions (e.g., *"Take two twice daily"*) vs. patient centred UMS labels. These labels contain simplified text, numeric characters instead of words to detail the dose, and "carriage returns" to place each dose on a separate line to clearly identify every time period a medicine is taken (e.g. *"Take 2 tablets in the morning and 2 tablets in the evening"*) | TE: Patients receiving the UMS labels consolidated their medicines into more times per day than those in receiving the standard labels, but no statistically significant difference was found. HL: 44.7% of the patients had limited health literacy. Subgroup analysis did not find any additional benefit of UMS labels in those with limited health literacy (Mean score 8.56 vs. 9.06, p = 0.514), but rather in those who said that they found instructions hard to understand (mean score 10.00 vs. 8.43, p = 0.019). | Specificity of dosage instruction, plain language and number of messages per instruction line | Use simplified text, numeric characteristics instead of words to detail dose, place each dose on a separate line to identify every time period a medicine should be taken. |

* Abbreviations: DLIs, drug label instructions; CI, confidence interval; HL, health literacy; NVS, Newest Vital Sign; REALM, Rapid Estimate of Adult Literacy in Medicine; RR, relative risk; SBSQ-D, Set of Brief Screening Questions in Dutch; TE, Textual Element; TOFHLA, Test Of Functional Health Literacy in Adults; UMS, Universal Medication Schedule.

** Publications based on the same study.

Instruments measuring health literacy were also identified. We checked if they only measured functional health literacy, or whether they measured interactive and/or critical health literacy skills as well.

As the studies were heterogeneous with respect to type of intervention, primary outcome and study design, pooling of the results was considered inappropriate [36]. We aimed to give a graphical impression of the potential effects of interventions and the influence of health literacy on improved comprehension. Effects and certainty were expressed as presented in the original studies as Relative Risk (RR) or Odds Ratio (OR) with corresponding confidence intervals. Two figures were plotted with the main quantitative outcomes (RR/OR), confidence intervals, number of included patients, and a description of the intervention for the effects of interventions on improved comprehension (Fig 3) and influence of health literacy on comprehension (Fig 4). The studies expressed their primary outcome differently (e.g., patients with correct understanding or misunderstanding), hence we recalculated, if necessary, the relative risks and odds ratios to the effects on improved comprehension. Similarly, we recalculated all comparisons for health literacy to the effects of higher levels of health literacy compared to lower levels of health literacy.

The questions in the JBI checklists for cross-sectional studies, experimental studies, qualitative studies, and randomized controlled trials are specified in the critical appraisal tools available on the website of the Joanna Briggs Institute [33].

## Results

### Selection of studies

A total of 434 records were identified in our literature search: PubMed (n = 177), SmartCat (n = 109), PsychINFO (n = 16), and Embase (n = 132). Records were eligible for inclusion if they met the following criteria: (1) studies which dealt with the comprehensibility of DLIs, and (2) actual studies which addressed textual elements of DLIs. A total of 60 studies remained for full-text review. Fig 1 presents a complete overview of the article search and review process. Finally, 30 publications addressing textual elements that affect the comprehension of DLIs were included in our review, of which four articles were merged into two studies in Table 3. The articles of Wolf et al. [17] and Davis et al. [26] were based on one study as well as the studies of Shiyanbola [34,35]. Therefore, 28 studies are mentioned in Table 3.

### Characteristics of the included studies

Table 3 presents a complete overview of the included studies. The 28 included studies used different research methods: in-person structured interviews (n = 19), surveys (n = 4), discussion groups (n = 3), and randomized controlled experiments (n = 2). The comprehension of DLIs was measured in participant groups varying from 21 to 845 participants. Dosage instructions were most often investigated (n = 16), followed by auxiliary instructions (n = 6), and a combination of instructions (n = 6). Also, most studies investigated DLIs of prescriptions (n = 22) or DLIs that could occur on both prescribed and over-the-counter drugs (n = 5). One study focused on the comprehension of over-the-counter DLIs (n = 1). The included studies used different methods to measure participants' comprehension of DLIs. In 16 studies, participants' comprehension was assessed by rephrasing the instructions in their own words (e.g., 'What do you think this auxiliary label is saying?), whereas four studies asked participants to demonstrate when and how many tablets they would take on a day. Four studies used multiple methods to measure participants' DLI comprehension (i.e., rephrasing the DLI in their own words as well as demonstrating when and how many tablets they would take). In four other studies patients' comprehension was assessed using a survey (n = 2), a discussion group (n = 1) and an

interview in which patients' perspectives on the formulation of DLIs were discussed (n = 1). Finally, in 19 studies participants' health literacy skills were assessed.

## Interventions and textual elements in DLIs

The following textual elements were studied: the specificity of dosage instructions (implicit vs. explicit formulation), plain language, wording of dose measurements, the number of messages per instruction line, the presentation of numbers (numerical vs. alphanumerical), DLIs in patients' native language, and the use of abbreviations. Fig 2 provides an overview of the frequency of textual elements in the included studies. Also, Table 4 provides an overview of the textual elements investigated in the included studies, and the quantitative data regarding the comprehensibility of DLIs.

In total, seven of the 28 studies investigated the effect of textual interventions in DLIs on participants' comprehension. Fig 3 gives an impression of these quantitative findings: four out of seven showed that interventions in DLIs significantly improved the comprehension, three studies found no effect.

## Specificity of dosage instructions

A total of 17 studies investigated the specificity dosage instructions: using implicit dosage intervals or explicit dosage intervals. Implicit dosage intervals only mention the frequency of intake (e.g., *'2 times daily'*), whereas explicit dosage intervals mention the moment of intake specified by the hour of intake, dayparts, or mealtime anchors (e.g., '*1 tablet in the morning and 1 tablet in the evening'*).

Misunderstanding was less frequent for explicit dosage instructions [9,11,36]. Correct interpretation of dosage instructions increased when providing dosage intervals specified by four distinct time periods (i.e., morning, noon, evening, and bedtime) [9,37,38,49] or hour of intake [36,37]. Only one study on administering liquid paediatric medication using implicit versus explicit dosage instructions concluded that there was no significant difference in patient comprehension [39]. Two studies [36,40] recommended explicit DLIs, however, without defining what implicit and explicit dosage intervals were.

## Plain language

In 15 studies, plain language is acknowledged as an important factor affecting the comprehensibility of DLIs [5,8,13,14,17,34,35,41–48]. These studies recommended instructions should be brief, clear, and concrete. Medical jargon should be avoided (*e.g.*, *'subcutaneously'*, *'inhalation'*) and complex words *(e.g., 'prolonged or excessive exposure'*, *'tablets')* should be substituted for simpler ones (e.g., *'under your skin'*, *'puff'*, *limit your time in the sun'*, *'pills')*

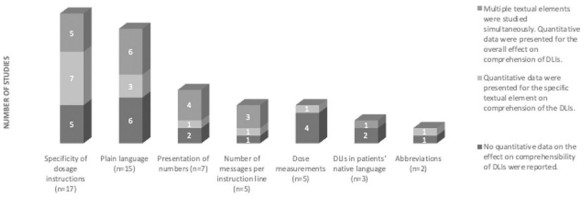

**Fig 2. The frequency of textual elements in the included studies (n), and the way in which the quantitative data were presented in the studies.**

**Table 4. The frequency of textual elements in the included studies, and, (if provided), an overview of the quantitative data on the effect of (combined) textual elements on the comprehensibility of the DLIs (n = 28).**

| Textual elements | Mentioned* in: | Quantitative data about the effect of (combined) textual elements on the comprehensibility of DLIs |
|---|---|---|
| **Specificity of dosage instructions, n = 17** | 1, 13, 15, 20, 21*** | No quantitative data presented. |
| | 2.** | Effects of combined textual elements were studied (ConcordantRx label). ConcordantRx label (four distinct time periods, plain language, lowercase and uppercase letters, and numeric characters) improved comprehension (RR: 1.25, 95% CI: 1.06–1.48). |
| | 3a.** | Misunderstanding was less frequent for instructions with explicit dosage instructions. |

Table for 3a.**:

| Drug label instruction | Correct understanding according to health literacy level (%) | | | p-value |
|---|---|---|---|---|
| | Low | marginal | adequate | |
| Take one tablet by mouth once each day | 86.7 | 87.6 | 94.7 | 0.032 |
| Take one tablet in the morning and one at 5 p.m. | 82.7 | 91.2 | 91.3 | 0.092 |
| Take two tablets by mouth twice daily | 70.7 | 84.1 | 89.4 | <0.001 |
| Take one teaspoonful by mouth three times daily | 58.7 | 65.5 | 82.6 | <0.001 |
| Take one tablet by mouth twice daily for seven days | 52.0 | 66.4 | 73.0 | <0.001 |

| Textual elements | Mentioned* in: | Quantitative data about the effect of (combined) textual elements on the comprehensibility of DLIs |
|---|---|---|
| | 3b. | Effects of combined elements were studied (Patient Centered Label, PCL). Better proper use at first exposure (76.9% vs. 70.1%, p = 0.06). Better proper use at 9 months (85.9% vs. 77.4%, p = 0.03). Subgroup analysis: effect significant for English-speaking patients (OR: 2.21, 95% CI: 1.13–4.31), not for Spanish speakers (OR: 1.19, 95% CI:0.63–2.24). PCL did not improve medication adherence. |
| | 17. | Rates of correct interpretation were lowest for instructions that depicted frequency in hourly intervals or the number of times of day (53% and 61%, respectively) and highest for those that used time periods (89%). |
| | 18. | Only 4 (0.93%) out of 429 prescriptions specifying daily frequency of dosage were misinterpreted, whereas 55 (77%) of the 71 prescriptions specifying hourly intervals were misinterpreted (RR: 83, 95% CI: 31–200). |
| | 19. | DLIs with daily terms were more frequently associated with correct comprehension than meal terms (rates of incorrect responses were 24.2% and 52.2% respectively). |
| | 23.** | Effects of combined elements were studied (Take-Wait-Stop label, TWS). 14% of the participants using the Take-Wait-Stop label exceeded the maximum dose. 31.8% of the participants using the Take-Wait-Stop label exceeded the maximum dose. Difference between TWS-label and standard label was significant (OR = 2.5, 95% CI 1.05–2.70). |
| | 24. | Effects of combined elements were studied (Patient Centered Label, PCL). PCLs specify explicit timing with standard intervals (morning, noon, evening, bedtime). PCL labels were more likely to be correctly interpreted than the standard instructions (ARR: 1.08, 95% CI: 0.98–1.18). |
| | 25. | Effects of implicit vs. explicit dosage intervals were studied Describe DLI Implicit dosage interval: 37.5% correct [OR: 1.00] Explicit dosage interval: 37.1% correct [OR: 1.01, 95% CI: 0.50–1.88] Demonstrate DLI Implicit dosage interval: 80.2% correct [OR: 1.00] Explicit dosage interval: 70.1% correct [OR: 0.53, 95% CI: 0.25–1.12] |
| | 26. | Effects of combined elements were studied (Patient Centered Label, PCL). DLIs with the PCL format were significantly more likely to be correctly interpreted compared to standard instructions (ARR: 1.33, 95% CI: 1.25–1.41, p<0.001) |
| | 27** | Effects of PCLs were studied. More proper use of drug regimens at first exposure (76.9% vs. 70.1%, p = 0.06) and at 9 months (85.9% vs. 77.4%, p = 0.03). |
| | 28. | Effects of Universal Medication Scheme (UMS) were studied. Those in the UMS group displayed better understanding of the prescription regimen than those in the usual care group, but this was not statistically significant. (Mean score 9.28 vs. 8.81, p = 0.135). Subgroup analysis: no additional benefit of UMS for patients with limited health literacy (mean score 8.56 vs 9.06, p = 0.514), but rather in those who said that they found instructions on tablets hard to understand (mean score 10.00 vs 8.43, p = 0.019). |

*(Continued)*

**Table 4.** (Continued)

| Textual elements | Mentioned* in: | Quantitative data about the effect of (combined) textual elements on the comprehensibility of DLIs |
|---|---|---|
| **Plain language, n = 15** | 4, 6, 8a, 8b, 9, 14*** | No quantitative data presented. |
| | 2** | Effects of combined elements were studied (ConcordantRx label). ConcordantRx label (four distinct time periods, plain language, lowercase and uppercase letters, and numeric characters) improved comprehension (RR: 1.25, 95% CI: 1.06–1.48). |
| | 3b.** | Effects of combined elements were studied (Patient Centered Label, PCL). 79% of patients could not recognize and pronounce 'antibiotic', 73% 'orally', 70% 'teaspoonful', 48% 'medication', 45% 'prescription', and 35% 'dose'. Poor word recognition may have contributed to patients misreading words on labels, such as 'tablespoon' instead of 'teaspoon'. This accounted for 9% of errors (n = 34). |
| | 5** | According to participants' interpretations of the newly developed labels, the labels understood best were those with the following indications: 'Avoid excessive sun exposure' (n = 61, 50.8%), 'Do not drink alcohol' (n = 49, 40.8%), and 'Take with plenty of water' (n = 49, 40.8%). |
| | 10** | Labels were at greater risk for being misunderstood if they included multiple instructions, or included unfamiliar terms. Rates of correct interpretation of the eight patient warning labels (PWL) ranged from 0% to 78.7%. Rates of comprehension among patients were the lowest for multiple-step PWLs (8.0%, 0%, and 5.3%). Specific PWLs were not understood by most patients. For example, 'For external use only' proved to be difficult for 90.7% of the participants. |
| | 11 | Simplified text warning labels (with and without icons) were studied. Simplified text + icon and simplified text only warnings more likely to be properly understood compared to standard warnings (92.1%, 90.6%, and 80.3% respectively; p<0.001). |
| | 12 | Comprehension of the enhanced text + icon label was significantly higher compared to both standard and enhanced text-only labels (icon vs standard: RR: 1.26; 95% CI: 1.04–1.53; icon vs enhanced text: RR:1.22; 95% CI: 1.02–1.46). |
| | 23** | Of the sample, 23% exceeded the maximum daily dose noted on the bottle and for this error type, there were statistically significant differences by study arm (standard label error rate = 31.8% vs. Take-Wait-Stop label error rate = 14%, p = .05). Those exposed to the standard label were 2.5 times more likely to exceed the recommended maximum daily dose (95% CI: 1.05, 7.70, p = .03). |
| | 26. | Effects of combined elements were studied (Patient Centered Label, PCL). DLIs with tPCL format were significantly more likely to be correctly interpreted compared to standard instructions (ARR: 1.33, 95% CI: 1.25–1.41, p<0.001) |
| | 28** | Effects of Universal Medication Scheme (UMS) were studied. Those in the UMS group displayed better understanding of the prescription regimen than those in the usual care group, but this was not statistically significant. (Mean score 9.28 vs 8.81, p = 0.135). Subgroup analysis: no additional benefit of UMS for patients with limited health literacy (mean score 8.56 vs 9.06, p = 0.514), but rather in those who said that they found instructions hard to understand (mean score 10.00 vs 8.43, p = 0.019). |
| **Dose measurements, n = 5** | 13, 14, 16, 21 *** | No quantitative data presented. |
| | 3a | Twenty-two percent of the patients with incorrect responses to the instructions, 'Take one teaspoonful by mouth three times daily', misinterpreted the dose as 'tablespoon' rather than 'teaspoon'. |

| Drug label instruction | Correct understanding according to health literacy level (%) | | | p-value |
|---|---|---|---|---|
| | Low | marginal | adequate | |
| Take one tablet by mouth once each day | 86.7 | 87.6 | 94.7 | 0.032 |
| Take one tablet in the morning and one at 5 p.m. | 82.7 | 91.2 | 91.3 | 0.092 |
| Take two tablets by mouth twice daily | 70.7 | 84.1 | 89.4 | <0.001 |
| Take one teaspoonful by mouth three times daily | 58.7 | 65.5 | 82.6 | <0.001 |
| Take one tablet by mouth twice daily for seven days | 52.0 | 66.4 | 73.0 | <0.001 |

**Table 4.** (Continued)

| Textual elements | Mentioned* in: | Quantitative data about the effect of (combined) textual elements on the comprehensibility of DLIs |
|---|---|---|
| **Number of messages per instruction line**, n = 5 | 7*** | No quantitative data presented. |
| | 5** | To make the new labels more comprehensible, the researchers used single-action directions, plain-language text, and explicit pictorial descriptions of the warning message.<br>According to participants' interpretations of the newly developed labels, the labels understood best were those with the following DLIs (single-action instructions): 'Avoid excessive sun exposure' (n = 61, 50.8%), 'Do not drink alcohol' (n = 49, 40.8%), and 'Take with plenty of water' (n = 49, 40.8%). |
| | 10 | Understanding of DLIs for liquid medication was studied.<br>Rates of comprehension were lowest for the three patient warning labels with multiple precautions or steps instructing proper use of medication (8.0%, 0%, and 5.3%). |
| | 23** | Effects of combined elements were studied (Take-Wait-Stop label, TWS).<br>14% of the participants using the Take-Wait-Stop label exceeded the maximum dose.<br>31.8% of the participants using the Take-Wait-Stop label exceeded the maximum dose. Difference between TWS-label and standard label was significant (OR: 2.5, 95% CI 1.05–2.70). |
| | 28** | Effects of Universal Medication Scheme (UMS) were studied.<br>Those in the UMS group displayed better understanding of the prescription regimen than those in the usual care group, but this was not statistically significant. (Mean score 9.28 vs. 8.81, p = 0.135). |
| **Presentation of numbers**, n = 7 | 1, 14*** | No quantitative data presented. |
| | 2** | Effects of combined elements were studied (ConcordantRx label).<br>ConcordantRx label (four distinct time periods, plain language, lowercase and uppercase letters, and numeric characters) improved comprehension (RR: 1.25, 95% CI: 1.06–1.48). |
| | 3a** | Mistakes were more common when the instructions consisted of several components with varying numerical information. |

| Drug label instruction | Correct understanding according to health literacy level (%) | | | p-value |
|---|---|---|---|---|
| | Low | marginal | adequate | |
| Take one tablet by mouth once each day | 86.7 | 87.6 | 94.7 | 0.032 |
| Take one tablet in the morning and one at 5 p.m. | 82.7 | 91.2 | 91.3 | 0.092 |
| Take two tablets by mouth twice daily | 70.7 | 84.1 | 89.4 | <0.001 |
| Take one teaspoonful by mouth three times daily | 58.7 | 65.5 | 82.6 | <0.001 |
| Take one tablet by mouth twice daily for seven days | 52.0 | 66.4 | 73.0 | <0.001 |

| Textual elements | Mentioned* in: | Quantitative data about the effect of (combined) textual elements on the comprehensibility of DLIs |
|---|---|---|
| | 3b. | Effects of combined elements were studied (Patient Centered Label, PCL).<br>Errors that appeared to be the result of label language were most prevalent on the instruction *'Take two tablets by mouth twice daily'*. The repetitiveness between dosage ('two') and frequency ('twice') often led to the common interpretation 'Take a pill twice a day', whereas dosage would go ignored. This was confirmed in the follow-up demonstration task, 'How many pills would you take in one day' with the common incorrect response of 'two' (72% of incorrect responses). |
| | 23** | Effects of combined elements were studied (Take-Wait-Stop label, TWS).<br>Of the sample, 23% exceeded the maximum daily dose noted on the bottle and for this error type, there were statistically significant differences by study arm (standard label error rate = 31.8% vs. Take-Wait-Stop label error rate = 14%, p = .05). Those exposed to the standard label were 2.5 times more likely to exceed the recommended maximum daily dose (95% CI: 1.05, 7.70, p = .03). |
| | 28** | Effects of Universal Medication Scheme (UMS) were studied.<br>Those in the UMS group displayed better understanding of the prescription regimen than those in the usual care group, but this was not statistically significant. (Mean score 9.28 vs. 8.81, p = 0.135).<br>Subgroup analysis: no additional benefit of UMS for patients with limited health literacy (mean score 8.56 vs. 9.06, p = 0.514), but rather in those who said that they found instructions on tablets hard to understand (mean score 10.00 vs. 8.43, p = 0.019). |

(*Continued*)

**Table 4.** (Continued)

| Textual elements | Mentioned* in: | Quantitative data about the effect of (combined) textual elements on the comprehensibility of DLIs |
|---|---|---|
| **DLIs in patients' native language, n = 3** | 1, 4*** | No quantitative data presented. |
| | 2** | Effects of combined elements were studied (ConcordantRx label).<br>Language concordantRx label improved comprehension (RR: 1.25, 95% CI: 1.06–1.48). |
| **Abbreviations, n = 2** | 6*** | No quantitative data presented. |
| | 22 | Effects of simplified text labels (with and without icons) were studied.<br>Simplified text labels, with and without patient-centred icons were better attended to by patients than standard labels (simplified text: AOR: 1.17, 95% CI: 1.02–1.36. |

*Numbers refer to the numbers used in Table 3.

** Quantitative data indicates understanding of several textual elements in DLIs.

*** No quantitative data on the understanding of the textual element(s) in DLIs.

Abbreviations: AOR, Adjusted Odds Ratio; ARR, Adjusted Relative Risk; CI, Confidence Interval; PCL, Patient-Centred Label; PWL, Patient Warning Label; OR, Odds Ratio; TWS, Take-Wait-Stop label; RR, Relative Risk; UMS, Universal Medication Scheme.

[8,14,45,46]. Four studies [13,14,42,44] recommended using plain language, however, without defining what plain language entails.

## Dose measurements

The formulation of dose measurements should be concrete and simple. Research showed words, such as 'teaspoon', 'tablespoon', 'dropperful', 'ml', and 'a small amount', are often misinterpreted due to participants' unfamiliarity with the terminology or found the terminology confusing [5,11,12,18,45,49]. However, the research did not provide recommendations for alternative formulations for dose measurements other than avoiding the above-mentioned examples.

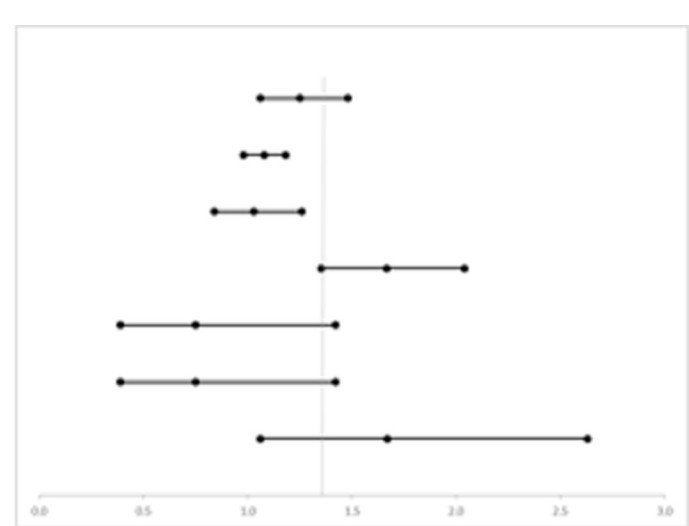

¹ Enhanced text (without icon) vs usual text.
² Study calculates misinterpretation of DLI. We represent the inversed data.
³ Improved comprehension of proper use at 9 month after intervention.
⁴ For drug adherence as outcome results were not significant (OR=1.59, 0.93-2.57, See Table 3).

**Fig 3. Relative risks and odds ratios of interventions in DLI on correct comprehension.**

## Number of messages per instruction line

Five studies found multi-step instructions (e.g., *'You should avoid prolonged or excessive exposure to direct and/or artificial sunlight'*) while taking this medication lead to more misinterpretations [14,26,42,46]. Participants became confused when interpreting the multi-step instructions or did not address all messages of the drug label. Placing each part of the instruction on a separate instruction line (i.e., '*carriage returns*') was proposed as an effective way to enhance comprehensibility [26] as well as the use of single step instructions (e.g. '*Take with food*') [14,27,44].

## Presentation of numbers

Four studies recommend the use of numeric over alphanumeric presentation of numbers in DLIs (e.g., *1* vs. *one*) [5,17,45,46]. However, fractions (i.e., ½) were better understood when presented alphanumerical (i.e., *half*) as the former lead to more confusion [45]. Research also showed DLIs containing fewer numbers (e.g., *'Take one tablet by mouth once each day'*) were better understood compared to instructions containing multiple numbers (*'Take one tablet by mouth twice daily for seven days'*).

## DLIs in patients' native language

Three studies recommended providing instructions in patients' native language [5,36,41]. Patients were more likely to demonstrate the correct dosing amount when receiving instructions in their native language instead of standard instructions [5,36,41].

## Abbreviations

Only one study compared patients' comprehension of the use of Latin abbreviations in DLIs (e.g., '*Take one tablet TID*') versus written-out instructions (e.g., '*Take one tablet three times day*') [50]. The Latin abbreviations were least understood compared to the written-out instructions. Also, two other studies recommend to avoid abbreviations (e.g., '*ml*') [43,45,50].

## Health literacy and drug label understanding

Participants' health literacy skills were measured in 19 studies, using the following measures: the REALM (n = 13), the NVS (n = 2), the revised REALM (n = 1), the Arabic Single Item Literacy Screener (n = 1), the Set of Brief Screening Questions (SBSQ) (n = 1), Short Assessment of Health Literacy for Spanish Adults (SAHLSA (n = 1) and the Test Of Functional Health Literacy in Adults (TOFHLA) (n = 1). One study used two measures (SAHLSA and REALM) depending on participants' native language [51]. All instruments measured functional health literacy.

In most studies the effectiveness of specific textual elements on the comprehensibility of DLIs for participants with limited health literacy skills was not measured separately, and, therefore cannot be described in this review. However, five studies compared DLI comprehension between people with lower and higher levels of health literacy. In these studies, health literacy was classified as low, marginal or adequate. A graphical impression of these comparisons is presented in Fig 4 (also see Table 4). For these five studies, the results showed that participants with adequate health literacy comprehended DLIs better than people with limited health literacy. Two out of five studies also showed participants with adequate health literacy understood DLIs better compared to participants with marginal health literacy. However, the three other studies found no difference between these groups.

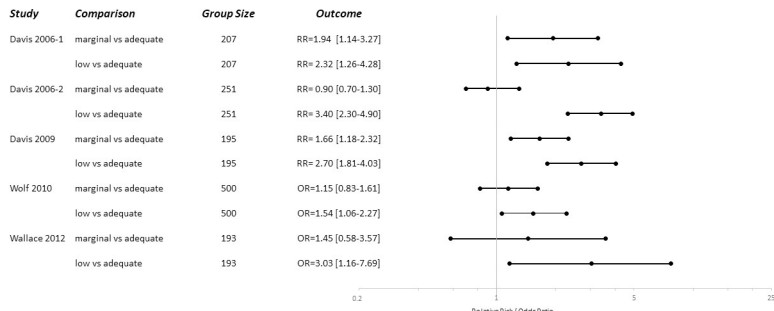

**Fig 4. Relative risks and odds ratios of misunderstanding of patients with marginal/low health literacy compared to patients with adequate health literacy.**

Two studies investigated the impact of textual DLI interventions on participants' comprehension and took respondents' health literacy into account. In 2010, Wolf et al. [13] investigated the effects of an enhanced label for auxiliary warnings with simplified text and, in addition, simplified icons. As described in Table 4, and shown in Fig 4, in all patients simplified text improved DLI comprehension. In participants with low health literacy, DLI comprehension was improved by the combination of simplified text and icons [OR 3.22, 95% CI 1.39–7.50]–but not for simplified text alone (see Fig 4). In 2016, Wolf et al. [38] found that the Patient Centred Label (PCL) led to slightly better use of drug regimens. The results were not significant for the entire population. However for the subgroup of participants with low health literacy, the intervention improved drug adherence (calculated by pill count) at nine months after the start of the intervention [OR = 5.08, 95% CI 1.15–22.37].

## Discussion

All studies in this review conclude that standard DLIs are too complex for patients which is attributed to their wording. Explicit time periods in dosage instructions, plain language, numbers in a numerical format, and providing DLIs in patients' native language contributed to improved comprehension. Correct interpretation of DLIs was hindered by multistep instructions per instruction line as well as using abbreviations and medical jargon. The included studies differ in the adopted research methods as well as their respondent population, which suggests this is a robust conclusion. Although health literacy was taken into account in a majority of the studies, none of them assessed the effectiveness of specific textual elements on patients' comprehension of DLIs.

The reviewed research is consistent about specifying dosage instructions and using plain language. Dosage instructions indicating more times daily use without specifying the time are often misunderstood (e.g., '*1 tablet 2 times a day*'). Using dayparts or mealtime anchors to specify the moment of intake facilitates patients' comprehension (e.g., '*1 tablet in the morning; 1 tablet in the evening*'). Future research should investigate whether dayparts are to be preferred over mealtime anchors, as meals are often skipped or consumed on different moments [52]. However, using time intervals to specify the moment of intake (e.g., '*every 8 hours*') should be avoided, as both Hanchak et al. and Davis et al. showed that about 75% of the participants misunderstood these instructions [9,53]. Regarding the use of plain language, the reviewed research recommends to avoid medical jargon and to substitute complex words for simpler ones. However, only few studies provide directions to simplify DLIs or provide an overview of alternative formulations for difficult words and/or medical jargon [5,8,13,41,45]. Moreover, the evidence on the wording of dose measurements, presentation of numbers,

presence of abbreviations, number of messages per instruction line as well as the use of DLIs in patients' native language is limited.

Due to the heterogeneity of the included studies with respect to primary outcome and study design pooling of the results was considered inappropriate. The effect of the individual textual elements on the comprehensibility of DLIs was not always presented quantitively, and relatively few studies compared the effect of optimized DLIs to standard DLIs. However, the included studies that allowed a quantitative comparison showed that textual interventions in DLIs are promising: optimized DLIs seem to have a positive impact on comprehensibility, especially in patients with limited health literacy. This might be explained by the fact that patients with limited health literacy experience more problems with the comprehension of DLIs, so there is more room for improvement in this population.

In 19 studies, participants' health literacy was measured with instruments focusing on functional skills. For example, the REALM measures whether participants can read and pronounce medical terms. However, being able to read and pronounce medical terms does not necessarily imply that patients are able to interpret medical terms in DLIs. Moreover, the reviewed research shows patients are able to rephrase DLIs in their own words, but experience difficulty in demonstrating proper drug use [5,46,47,49,51]. Therefore, it is likely that beside functional skills, interactive and critical skills also play a role when interpreting and applying DLIs. Future research should therefore incorporate multiple health literacy measures in order to cover these aspects of health literacy as well as multiple comprehension measures (i.e., reading, interpreting, and demonstrating).

At the start of a new drug therapy, patients usually receive DLIs combined with spoken information from prescriber and written information in patient leaflets. However, approximately 40 to 80 per cent of the information during patient-physician encounters is forgotten or remembered inaccurately [2–4]. Unlike patient information leaflets, DLIs present only the most essential information on drug use and are likely to be the last information source patients read before taking their drugs [8]. Hence, DLIs should serve as an independent text which should be comprehensible for all patients. This systematic review shows specifying dosage instructions and using plain language may facilitate patients' comprehension.

## Strengths and limitations

As far as we know, this is the first systematic review investigating the relation between textual elements in DLIs and patients' comprehension. A strength of this review is the focus on DLIs instead of focussing on other types of health information sources. Other strengths are the focus on textual elements in DLIs and (if possible), and the inclusion of patients' health literacy in the data analysis.

A first limitation of this review is a consequence of our focus on textual elements: we did not study other elements, such as icons, that may impact DLI comprehensibility. Additional elements, such as icons or pictograms, could facilitate the comprehension of DLIs. For example, a systematic review of Sletvold et al. [54] showed pharmaceutical pictograms combined with written/oral information are useful for patients that are normally at risk for non-adherence. Also, Katz et al. [55] concluded a combination of pictorial aids and textual information facilitate patients' comprehension of medication instructions compared to pictorial aids only. Therefore, future research should focus on the effectiveness of both textual and visual elements in DLIs on patients' comprehension.

Another limitation is that most studies in this review were not performed in a clinical setting and used hypothetical drug labels. Participants had to interpret instructions of drugs they did not use themselves. Therefore, the occurrence of misinterpreting DLIs might be

underestimated. This especially goes for patients following multiple drugs regimens and patients with limited health literacy. Future research should therefore investigate patients' understanding of instructions of their own medications (cf. [56]) to increase the ecological validity of research on the comprehensibility of DLIs.

A final limitation is the heterogeneity of the included studies in this review and the differences in data presented on the comprehension of DLIs. This makes it impossible to quantify the effectiveness of textual interventions in general, and to identify the contribution of specific textual elements. Although the data in this review illustrate that (decreased) comprehensibility of DLIs is a substantial problem, especially in patients with limited health literacy, it would be valuable to have pooled data on how (optimized) DLIs influence patients' comprehension, and other outcomes, such as adherence to treatment regimens and even clinical outcomes. We would welcome such comparisons, however, we believe heterogeneity will also be problematic in future evaluations as interventions directed at improved drug use will always be context specific, with multiple factors contributing to the primary outcome.

### Implications for future research

Our results underline that the wording of DLIs impacts patients' comprehension. The textual elements found in this review can be used to improve existing drug labels instructions, such as specifying dosage instructions and using plain language. Although health literacy skills are a known predictor of patients' comprehension of DLIs, little is known about the effectiveness of specific textual interventions for patients with marginal or limited literacy skills. Therefore, future research should focus on the effectiveness of specific textual interventions and should include patients' health literacy in the research design. Moreover, the patients' comprehension of DLIs should be tested in a clinical setting using the DLIs of patients' own medication.

### Implications for practice

Based on this review, DLIs could be optimized by specifying dosage instructions and using plain language. In many countries, prescribing and dispensing is supported by healthcare information systems [55] which contains a centrally maintained table for all available dosing instructions on DLI with corresponding codes. When processing a prescription, the health care professional will use these predefined DLI codes in order to automatically print the instructions on the drug label. Depending on the health care context, optimized DLIs can be implemented by adapting the associated codes in these information systems. Also, guidelines for the composition of comprehensible DLIs can be developed to support prescribing and dispensing professionals.

### Conclusion

The present systematic review documents which textual elements have been investigated to facilitate patients' comprehension of DLIs: specifying dosage instructions and using plain language are promising ways to increase DLI comprehensibility. Especially, patients with limited health literacy might benefit from optimized DLIs. However, the heterogeneity in study design, the textual interventions, and outcomes measured, prevents us from conclusively asserting that specific textual elements are effective in improving patients' comprehension of DLIs.

### Supporting information

**S1 Checklist. PRISMA 2009 checklist.**
(DOC)

## Acknowledgments

We would like to thank Reza Bandraz for his contribution to the study selection.

## Author Contributions

**Conceptualization:** Charlotte Miriam Joyce van Hooijdonk.

**Funding acquisition:** Charlotte Miriam Joyce van Hooijdonk, Liset van Dijk, Jany Rademakers, Sander Diederik Borgsteede.

**Methodology:** Ekram Maghroudi, Charlotte Miriam Joyce van Hooijdonk, Heidi van de Bruinhorst, Liset van Dijk, Jany Rademakers, Sander Diederik Borgsteede.

**Supervision:** Charlotte Miriam Joyce van Hooijdonk, Sander Diederik Borgsteede.

**Writing – original draft:** Ekram Maghroudi, Charlotte Miriam Joyce van Hooijdonk, Sander Diederik Borgsteede.

**Writing – review & editing:** Ekram Maghroudi, Charlotte Miriam Joyce van Hooijdonk, Heidi van de Bruinhorst, Liset van Dijk, Jany Rademakers, Sander Diederik Borgsteede.

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
