## [Decision Letter · Decision Letter 0]

27 Mar 2020

PONE-D-19-35643

The impact of textual elements on the comprehensibility of drug label instructions: a systematic review

PLOS ONE

Dear mrs. Maghroudi,

Thank you for submitting your manuscript to PLOS ONE. After careful consideration, we feel that it has merit but does not fully meet PLOS ONE’s publication criteria as it currently stands. Therefore, we invite you to submit a revised version of the manuscript that addresses the points raised during the review process.

We would appreciate receiving your revised manuscript by May 11 2020 11:59PM. To enhance the reproducibility of your results, we recommend that if applicable you deposit your laboratory protocols in protocols.io, where a protocol can be assigned its own identifier (DOI) such that it can be cited independently in the future. For instructions see: http://journals.plos.org/plosone/s/submission-guidelines#loc-laboratory-protocols

We look forward to receiving your revised manuscript.

Kind regards,

Sandra Laing Gillam, Ph.D

Academic Editor

PLOS ONE

Additional Editor Comments (if provided):

Dear Dr. Maghroudi,

Thank you for submitting your manuscript, “The impact of textual elements on the comprehensibility of drug label instructions: a systematic review. The topic is a very important one and has the potential to contribute to the literature in important ways. However, in order for the manuscript to be published in PLOS ONE it must be written in an intelligible fashion in standard English. The journal does not copyedit manuscripts so articles must be clear, correct and unambiguous with no grammatical, wording or syntactic errors. The manuscript, as written is difficult to read because it does not meet this minimum requirement to the extent that it the reviewer and I were not entirely sure we could interpret the methodology, analysis and interpretation of the findings. As suggested by the reviewer, it would be helpful to have a proofreader revise the manuscript before resubmitting.

One very noticeable problem with the data is related to a failure to differentiate between “health literacy” specifically and “literacy” in general. As the reviewer points out, these are two entirely different constructs and as such, should be treated as different variables.

The review of the literature and rationale for the project should set up the analysis. Because there are essentially two entirely different constructs as outcome variables, the literature review would need to be revised to address them specifically. In addition, there are other factors that need to be considered and included in the literature review, such as stress variables as suggested by the reviewer. The reviewer had some very helpful comments for the authors to use to make the analysis and results sections more clear and comprehensive, so I would encourage the authors to make use of those suggestions. Please include full names and then acronyms so that readers do not become confused. With respect to effect sizes, there are a number of traditional and nontraditional ways to calculate effect sizes ranging from Cohen’s d to Standard Mean Difference (SMD), so I would encourage the authors to consult with a statistician to determine if there are ways to represent the findings in a way to evaluate participant responses in a meaningful way.

I hope you will address the issues we have identified and resubmit to this journal or another journal of your choosing.

Journal Requirements:

2. Please include your tables as part of your main manuscript and remove the individual files. Please note that supplementary tables (should remain/ be uploaded) as separate "supporting information" files

4. We note you have included a table to which you do not refer in the text of your manuscript. Please ensure that you refer to Table 3 in your text; if accepted, production will need this reference to link the reader to the Table.

5. Thank you for stating the following in the Funding details of your manuscript:

"This work was supported by ZonMw, under Grant 848022004."

"The funders had no role in study design, data collection and analysis, decision to publish, or preparation of the manuscript. "

Reviewers' comments:

Reviewer's Responses to Questions

**Comments to the Author**

1. Is the manuscript technically sound, and do the data support the conclusions?

Reviewer #1: Partly

2. Has the statistical analysis been performed appropriately and rigorously? 

Reviewer #1: N/A

3. Have the authors made all data underlying the findings in their manuscript fully available?

Reviewer #1: Yes

4. Is the manuscript presented in an intelligible fashion and written in standard English?

Reviewer #1: No

5. Review Comments to the Author

Reviewer #1: This manuscript is a systematic review/meta-analysis of previously published works on drug label instructions (DLI). The authors specifically asked whether textual elements of the labels made affect how well people understand them (and, presumably, follow the instructions).

I would make the following overall recommendations to the authors:

1. This manuscript is difficult to read owing to many atypical and/or erroneous lexical choices. In fact, it took so much effort to read the manuscript that I fear that I might have missed questions and details to help the authors improve the paper. I would recommend that, before resubmission to this or another journal, the authors ask someone who is not affiliated with the manuscript to act as a proofreader.

2. Several of the studies deal with literacy (i.e., “reading level,” e.g., Davis et al., 2006) rather than “health literacy.” I believe that the authors should separate these two constructs, re-assign those studies, and rework all sections of the paper to include "health literacy" and "literacy" as different constructs. Not understanding health, healthcare, and treatment protocols is a very different problem from not having the reading skill to comprehend the meaning of “take two pills twice per day.” I suspect that this problem might have native language vs. language of label as a confound, as many people are differentially literate in different languages they speak, in which case the authors would need to consider that separately.

3. A smaller detail: there is a large literature demonstrating that patients forget or misremember as much as 75% of what healthcare professionals say during appointments; I’m surprised not to see a single citation for this. (Citations in paragraph 1, pp 4 refer to written instructions.) Many of these studies also include health-related stress variables which would strengthen the discussion on pp 17.

In addition, I have some specific methodological questions. The instructions I was given from PLOS ONE indicated that I was to review this manuscript for technical aspects of the study, paying close attention to the specificity and detail used in describing the scientific method and results. Based on these instructions, I have the following comments.

1. The inclusion of the PRISMA checklist and associated files makes it very clear that the method used for this study was systematic. I did have a difficult time understanding Table 2, however. I believe, based on the text on page 9 (15 of the pdf), that Table 2 refers to assessment of methodological quality based on an assessment from the Joanna Briggs Institute, and labels the questions used as Q1, Q2, Q3…Q13. It would be much easier for the reader who is not intimately familiar with this checklist if the topic areas of the questions were included on the table along with the question numbers.

2. On pp 8, the authors use text to describe some of the measures they took to maintain internal reliability and validity. There are some details here that I don’t understand:

a. Why were 20% of the studies independently assessed by title and abstract, and then fully assessed? Were these different assessments? What was inter-assessor agreement during this process? If I understand correctly, after this process, the remaining 80% of the studies were assessed by only one person and no inter-assessor reliability was calculated. It would be good to know what the initial agreement was to understand why the bulk of the studies were assessed without further reliability measures.

b. How did the authors determine that papers were “obvious duplicates?” Are these papers that were listed in multiple databases, or where the journal listing and the funding listing were retrieved separately?

c. Did both EM and HvB complete the JBI assessment for all of the studies, or did one do only 30% as in the preceding “Data Classification and Analysis” section?

3. pp 14: What do the following stand for: REALM, NVS-d, SBSQ-d, SAHLSA, TOFHLA?

4. I am accustomed to seeing meta-analyses that include effect sizes. Effect size analysis might not make statistical sense in this case, but I do think the paper would be stronger if there were a way to see graphically the group sizes and proportion of participant responses indicating concern for each item. E.g., for specificity of dosage instructions, what proportion of the participants in each listed articles reported that implicit wording was helpful. (I think this would involve including article 26 in this group). This would change table 4 considerably; perhaps Table 4 would become several figures.

6. PLOS authors have the option to publish the peer review history of their article (what does this mean?). If published, this will include your full peer review and any attached files.

Reviewer #1: No

---

## [Author Response · Author response to Decision Letter 0]

13 May 2020

Response to reviewers

Maghroudi et al. The impact of textual elements on the comprehensibility of drug label instructions (DLIs): a systematic review.

We would like to thank the reviewer and the Editor for their valuable comments, and the possibilities to improve the paper considerably. As first major improvement, we have distinguished the concepts literacy and (functional) health literacy, and describe how these concepts were used in the individual studies. Second, we summarized the most important quantitative data in three figures that we added, describing (1) an overview of the effects of intervention studies between intervention and control groups, (2) an overview of differences found between people with adequate versus marginal/low health literacy, and (3) the quantitative findings concerning the textual elements that were investigated in the studies. However, given the heterogeneity of the results, the main part of this review remains a qualitative synthesis. We describe the process more clearly in our Methods section. The third major improvement we have made was rewriting the language and grammar of the paper after external proofreading. Added with the other comments, we believe we were able to improve the manuscript considerably, and are open to discuss further improvements.

We would like to mention a modification in the PRISMA checklist: question #5 protocol and registration changed from yes to no, due to the situation that PROSPERO did not perform an external review of our protocol and will not publish our protocol for the systematic review. 

Before starting our review, we submitted our protocol to PROSPERO for external review and publication (number 128532). This is the protocol we used for this manuscript. PROSPERO mentioned they were unable to review all protocols that were submitted timely (as they received many protocols), and gave priority to protocols from the UK. Given our own timelines, we decided to start our data-extraction, simultaneously during the review process. The protocol was still under review at the moment of submission to PLOS ONE.

When questioning PROSPERO about the status of their review process, we mentioned we had started our data-extraction and analysis. Given the workload and waiting time for review of new protocols, PROSPERO decided not to review, and publish this protocol (30-03-2020). 

* Detailled responses in the Response to Reviewers

---

## [Decision Letter · Decision Letter 1]

18 Nov 2020

PONE-D-19-35643R1

The impact of textual elements on the comprehensibility of drug label instructions (DLIs): a systematic review

PLOS ONE

Dear Dr. Maghroudi,

Thank you for submitting your manuscript to PLOS ONE. After careful consideration, we feel that it has merit but does not fully meet PLOS ONE’s publication criteria as it currently stands. Therefore, we invite you to submit a revised version of the manuscript that addresses the points raised during the review process.

We look forward to receiving your revised manuscript.

Kind regards,

Sandra Laing Gillam, Ph.D

Academic Editor

PLOS ONE

Additional Editor Comments (if provided):

Dear Dr. Maghroudi,

Thank you for re-submitting your manuscript, “The impact of textual elements on the comprehensibility of drug label instructions: a systematic review. I apologize for the length of time it has taken for a decision to be made on your manuscript. We had a very difficult time finding two reviewers who would agree to review your resubmission.

I appreciate the attention that you and your co-authors have given to revising the paper. While the writing is much better in this revision, there remain a number of organizational, typographical, spelling, grammatical and syntactic errors related to the use of English; as well as a number of issues with vague and ambiguous descriptions of terms, outcome measures, and effect sizes. These issues contribute to difficulty in understanding the results and the interpretation of the results in the discussion section. Both reviewers highlight these issues in their comments.

The discussion section often departs to topics and/or conclusions that are not directly supported in the data. The reviewers noted this in their comments and I agree that refocusing the discussion to only address the interpretation of the data that is presented would greatly improve the paper.

One reviewer suggested the paper be rejected, however, I feel that the manuscript has potential, therefore, I am recommending a major revision. I urge the authors to attend carefully to all of the suggestions made by the reviewers if they should choose to resubmit the paper to PLOS ONE. Both reviewers give detailed, clear and valid suggestions that should improve the manuscript. Given the difficulty we had in recruiting reviewers for the manuscript, I am recommending that this be the final opportunity for revision.

Thank you for considering PLOS ONE as an outlet for your research. If you choose to resubmit, I will do my best to ensure that the review is completed in a timely manner, however, I cannot guarantee a positive outcome.

Sincerely,

Sandra Laing Gillam

Reviewers' comments:

Reviewer's Responses to Questions

**Comments to the Author**

1. If the authors have adequately addressed your comments raised in a previous round of review and you feel that this manuscript is now acceptable for publication, you may indicate that here to bypass the “Comments to the Author” section, enter your conflict of interest statement in the “Confidential to Editor” section, and submit your "Accept" recommendation.

Reviewer #2: (No Response)

Reviewer #3: (No Response)

2. Is the manuscript technically sound, and do the data support the conclusions?

Reviewer #2: Partly

Reviewer #3: Partly

3. Has the statistical analysis been performed appropriately and rigorously? 

Reviewer #2: N/A

Reviewer #3: No

4. Have the authors made all data underlying the findings in their manuscript fully available?

Reviewer #2: Yes

Reviewer #3: Yes

5. Is the manuscript presented in an intelligible fashion and written in standard English?

Reviewer #2: No

Reviewer #3: Yes

6. Review Comments to the Author

Reviewer #2: The authors report the results of a systematic review aiming to understand how textual elements impact the understanding of Drug Label Instructions (DFI). The key finding seems to be that DFIs are often too complex for people to understand, which is an important but relatively well-documented issue. It is clear the authors have taken great care to use a systematic method of review and they have summarized the information thoroughly. This is not an easy task given the heterogeneity of the studies. Given the obvious amount of effort put into this study, I regret that I don’t have a more favorable reaction to the manuscript itself. I provide the following comments in the hope that they may be of utility to the authors.

Clarity and Conclusions

To start, the manuscript would benefit from additional copyediting for typographical and grammatical issues.

At various points, I found the manuscript difficult to follow and the synthesis of the findings lacking. While it seems that much care was taken to conduct a careful review and summary of the papers, the resulting synthesis and conclusions are less compelling.

For example, the Wording section on pgs. 24-25 states that complexity is shown to be problematic; however, it’s not clear what specific ways to simplify would be helpful. In other words, complexity is stated to be an issue, but it’s not clear what a solution or effective simplification interventions have been shown to help. Similarly, the Dose Measurements section on pg. 25 doesn’t indicate alternative words that would be preferable to those discouraged. Do these prior studies not offer any substantive recommendations on how to improve things?

Another concern is that the Discussion section sometimes diverges from the actual findings into relatively speculative claims (e.g., suggesting that meal intervals not be used). While the authors acknowledge this has not been investigated specifically, it doesn’t necessarily seem to follow from the findings of this review. Further, the Implications for Research and Implications for Practice sections also seem to contain content that is only tangentially related to these topics and disconnected from the findings.

While limiting the scope of the research is necessary due to practical limitations, excluding work on visual elements of labels may be problematic. Indeed, there is a substantial amount of work examining how visual elements (e.g., icons) can be used to improve comprehension and other outcomes among lower (health) literacy groups. This could be one reason the review did not find many studies specifically examining textual interventions aimed at improving outcomes among individuals of lower literacy. In other words, because of the challenges that low literacy presents to processing textual information, researchers have likely looked elsewhere to graphics, icons, or other visual elements as solutions.

With so much of the paper focused on health literacy, it would also be helpful to explain the specific measures in more detail. For example, the REALM essentially measures an individual’s familiarity with a list of various medical terms and thus seems inherently linked to someone’s ability to interpret more complex terms on a medication label. How do the other measures compare to this?

It appears that the vast majority of studies mentioned focus on medications that are dispensed via a health care professional (e.g., pharmacist or doctor). Are studies of drugs that can be accessed without an intermediary (e.g., over-the-counter drugs in the United States) also included?

Best wishes to the authors as they continue this research.

Reviewer #3: This is an interesting review of the impact of drug label instructions on comprehensibility.

I think the authors have done a reasonable job of responding to the reviewer and editor comments.

Overall the writing is clear and easy to follow.

There are, however, a couple of items that need further clarification.

1. The authors don't define "comprehension" in the manuscript, but it is important to do so. The studies included in the review will have measured "comprehension" in different ways. For example some studies will ask participants to verbally describe what they will do, others will require participants to demonstrate (in some manner) what they will do (i.e. to demonstrate how many tablets they would take and when) and others will require both. This should be clarified throughout

2. Like the editorial comments, I think there are ways the authors could estimate effect sizes despite the different approaches and measures used in the studies. It is not clear that the authors are referring to heterogeneity in the statistical sense when offering that as a reason against combining studies with quantitative measures. At minimum, the way the current figures showing RR and OR for various groups needs to be more clearly described and defended.

3. More examples of explicit v 'implicit' dose instructions and "wording of dose measurements" is needed.

4. The statements provided in the results section are often not sufficiently specific

- Line 260 "DLIs with explicit instructions are effective" at what?

- Line 264 information about "wording" is not sufficiently clear. Examples would help

- Line 281 placing each part of the instruction on a separate line was "proposed as an effective way": is there data to support this?

- Line 288 presentation of numbers is "preferred": the review is focused on 'comprehension' is there data on comprehension?

- Line 299 native language is "recommended": is there data to support this

- Line 318: "a negative relation between comprehension of DLI and health literacy" suggest a **negative correlation**, but this is not what you have found.

- Line 318 the information on health literacy should better recognise that many studies have found benefits of textual components of DLI improving comprehension for people with both low and high health literacy (though a larger effect in people with low health literacy)

5. The lack of specificity in the results makes it hard to say that the key findings as presented in the first paragraph in the Discussion have been demonstrated.

6. Line 356 it is not clear what the authors are including in "plain language"

7. Line 358 you don't appear to have provided clear evidence regarding the use of "dayparts" as opposed to alternative ways to specify the dose. Please clarify.

7. PLOS authors have the option to publish the peer review history of their article (what does this mean?). If published, this will include your full peer review and any attached files.

Reviewer #2: No

Reviewer #3: **Yes: **Adam La Caze

---

## [Author Response · Author response to Decision Letter 1]

17 Dec 2020

We would like to thank the Reviewers and the Editor for their valuable comments and suggestions, and the possibilities to improve the paper considerably. In our response we explain point-wise how we improved the paper. We would like to mention the 5 major improvements:

(1) The second author (CvH) has extensively copy-edited the manuscript. This period, the first (and corresponding) author had little time available to take the lead in this extensive revision as she was responsible for intensive (online) education for nurses about health communication. CvH lead and drafted the revision. We believe that a shared first authorship is the best way to value the contributions of EM and CvH. 

(2) We have explained in more detail what is meant by ‘comprehension of DLIs’ throughout the manuscript. 

(3) We explain in the Introduction and Discussion what is measured by different health literacy instruments and how this is related to the comprehension of DLIs. 

(4) All recommendations on the use of textual elements are accompanied with examples in Table 3 and in the Results section. 

(5) In the Discussion section we synthesize the key findings of our review and only address the interpretation of the data presented. 

Added with the other comments, we believe we addressed all issues, and are open to discuss further improvements.

*Detailed responses can be found in the document 'Response to the Reviewers'.

---

## [Decision Letter · Decision Letter 2]

16 Mar 2021

PONE-D-19-35643R2

The impact of textual elements on the comprehensibility of drug label instructions (DLIs): a systematic review

PLOS ONE

Dear Dr. Maghroudi,

Thank you for submitting your manuscript to PLOS ONE. After careful consideration, we feel that it has merit but does not fully meet PLOS ONE’s publication criteria as it currently stands. Therefore, we invite you to submit a revised version of the manuscript that addresses the points raised during the review process.

We look forward to receiving your revised manuscript.

Kind regards,

Yen-Ming Huang, PhD

Academic Editor

PLOS ONE

Journal Requirements:

Additional Editor Comments (if provided):

Thank you for your submission to PLOS One. Your paper has now been reviewed by 2 reviewers. As you will see from their comments below, the reviewers found your paper interesting and relevant for the journal. However, they have a number of concerns and critical comments, but also suggestions on how to improve the manuscript.

You should address all comments and revise according to them or clearly state if you have another opinion.

We look forward to receive a revised manuscript for re-evaluation.

Reviewers' comments:

Reviewer's Responses to Questions

**Comments to the Author**

1. If the authors have adequately addressed your comments raised in a previous round of review and you feel that this manuscript is now acceptable for publication, you may indicate that here to bypass the “Comments to the Author” section, enter your conflict of interest statement in the “Confidential to Editor” section, and submit your "Accept" recommendation.

Reviewer #2: (No Response)

Reviewer #4: (No Response)

2. Is the manuscript technically sound, and do the data support the conclusions?

Reviewer #2: Yes

Reviewer #4: Partly

3. Has the statistical analysis been performed appropriately and rigorously? 

Reviewer #2: Yes

Reviewer #4: I Don't Know

4. Have the authors made all data underlying the findings in their manuscript fully available?

Reviewer #2: Yes

Reviewer #4: Yes

5. Is the manuscript presented in an intelligible fashion and written in standard English?

Reviewer #2: Yes

Reviewer #4: Yes

6. Review Comments to the Author

Reviewer #2: I’m delighted to say that I found the revised manuscript substantially improved and I appreciate the authors’ careful attention to the previous comments. In particular, I found the manuscript much easier to read overall and the summary of results clearer and more directly linked to the discussion/conclusions. At this point, I suggest the authors consider the following minor issues:

Line 48 – Consider changing “whom” to “which”.

Line 160-161 - Might add some text to clarify the following statement: “Twenty per cent of the identified titles and abstracts were reviewed by the two independent reviewers (EM and HB).” Does this indicate a sample was drawn to make the workload more manageable?

Line 242 – summarize inclusion criteria or remind the reader of where this was discussed previously.

Line 325 – replace “did no” with “did not”.

Line 388 – nine months.

Line 406 – “to specify the” instead of “to specify to”.

Line 464 – should it be drug “regimen” instead of “regime”?

Lines 492-494 – I didn’t understand the following recommendation: “Depending on the health care context, DLIs can be optimized by adapting the associated codes in these information systems.”

Table 2 – Consider adding a footnote under table 2 that lists the JBI checklist questions.

Figure 3 – should the vertical line in the figure be at 1.0 on the x-axis?

Reviewer #4: The manuscript is well written and easy to understand. This reviewer notes several areas in which the manuscript can be strengthened.

Although the methods seemed reasonable, I was not sure what the actual research question or purpose was.

There is already a lot done in this area, and there were several references with studies on the relationship between drug label info and FHL that were not even referenced. That makes the whole search strategy curious.

The implications for creating a new label with some key points has been in existence since almost a decade so I am unsure what new knowledge has been added here.

7. PLOS authors have the option to publish the peer review history of their article (what does this mean?). If published, this will include your full peer review and any attached files.

Reviewer #2: No

Reviewer #4: No

---

## [Author Response · Author response to Decision Letter 2]

1 Apr 2021

We would like to thank the Reviewers and the Editor for their valuable comments and suggestions, and the possibility to improve the paper. In our response we explain point-wise how we improved the paper. We would like to mention four improvements:

(1) Adjustments to the manuscript have been made based on the minor textual suggestions of Reviewer #2. 

(2) We clarify the statement about how the two reviewers decided to exclude studies based on the titles and abstract. To determine the reliability of this decision, twenty per cent of the sample was double coded by the reviewers, after which the Cohen’s kappa was computed. 

(3) We explain in the Discussion what is meant by our recommendation about implementing optimized DLI’s in health information systems.

(4) The aim of the paper is emphasized in the Abstract. Also, a research question is added in the Introduction.

Added with the other comments, we believe we addressed all issues, and are open to discuss further improvements. Please contact us, if there remain any issues with respect to this submission.

---

## [Editor Report · Decision Letter 3]

5 Apr 2021

The impact of textual elements on the comprehensibility of drug label instructions (DLIs): a systematic review

PONE-D-19-35643R3

Dear Dr. Maghroudi,

We’re pleased to inform you that your manuscript has been judged scientifically suitable for publication and will be formally accepted for publication once it meets all outstanding technical requirements.

Kind regards,

Yen-Ming Huang, PhD

Academic Editor

PLOS ONE

---

## [Editor Report · Acceptance letter]

22 Apr 2021

PONE-D-19-35643R3 

The impact of textual elements on the comprehensibility of drug label instructions (DLIs): a systematic review 

Dear Dr. Maghroudi:

I'm pleased to inform you that your manuscript has been deemed suitable for publication in PLOS ONE. Congratulations! Your manuscript is now with our production department. 

Kind regards, 

on behalf of

Dr. Yen-Ming Huang 

Academic Editor

PLOS ONE